# LEGO-PROVER: NEURAL THEOREM PROVING WITH GROWING LIBRARIES

**Haiming Wang**[1*]   **Huajian Xin**[1*]   **Chuanyang Zheng**[3]   **Zhengying Liu**[2†]
**Qingxing Cao**[1]   **Yinya Huang**[4]   **Jing Xiong**[1]   **Han Shi**[2]   **Enze Xie**[2]   **Jian Yin**[1†]
**Zhenguo Li**[2]   **Xiaodan Liang**[1,5,6†]
[1]Sun Yat-sen University   [2]Huawei Noah's Ark Lab   [3]The Chinese University of Hong Kong
[4]City University of Hong Kong   [5]MBZUAI   [6]DarkMatter AI Research
{wanghm39, xinhj, caoqx, xiongj69, issjyin}@mail2.sysu.edu.cn, cyzheng21@cse.cuhk.edu.hk,
{liuzhengying2, xie.enze, shi.han, Li.Zhenguo}@huawei.com, yinya.huang@hotmail.com,
xdliang328@gmail.com

## ABSTRACT

Despite the success of large language models (LLMs), the task of theorem proving still remains one of the hardest reasoning tasks that is far from being fully solved. Prior methods using language models have demonstrated promising results, but they still struggle to prove even middle school level theorems. One common limitation of these methods is that they assume a fixed theorem library during the whole theorem proving process. However, as we all know, creating new useful theorems or even new theories is not only helpful but crucial and necessary for advancing mathematics and proving harder and deeper results. In this work, we present LEGO-Prover, which employs a growing skill library containing verified lemmas as skills to augment the capability of LLMs used in theorem proving. By constructing the proof modularly, LEGO-Prover enables LLMs to utilize existing skills retrieved from the library and to create new skills during the proving process. These skills are further evolved (by prompting an LLM) to enrich the library on another scale. Modular and reusable skills are constantly added to the library to enable tackling increasingly intricate mathematical problems. Moreover, the learned library further bridges the gap between human proofs and formal proofs by making it easier to impute missing steps. LEGO-Prover advances the state-of-the-art pass rate on miniF2F-valid (48.0% to 57.0%) and miniF2F-test (45.5% to 50.0%). During the proving process, LEGO-Prover also generates over 20,000 skills (theorems/lemmas) and adds them to the growing library. Our ablation study indicates that these newly added skills are indeed helpful for proving theorems, resulting in a 4.9% improvement in success rate [1].

## 1 INTRODUCTION

The automation of formal reasoning tasks, such as theorem proving and mathematical proof formalization, represents a formidable challenge and an active area of research within the domain of artificial intelligence (Polu & Sutskever, 2020a; Han et al., 2022; Jiang et al., 2022a; First et al., 2023; Bansal et al., 2019; Lample et al., 2022; Jiang et al., 2022b; 2021; Zhao et al., 2023; Yang et al., 2023; Wang et al., 2023b; Liu et al., 2023; Xiong et al., 2023; 2024; Huang et al., 2024). The process of formalizing mathematical proofs typically relies on human experts to transcribe intricate mathematical concepts into structured formal languages verifiable by interactive theorem prover like Lean (de Moura et al., 2015) or Isabelle (Paulson, 1994). This process, while robust, is often labor-intensive and demands a high level of expertise.

In the past few years, large language models (LLMs) have emerged as a promising avenue, with their capacity to process and produce human-like text, opening doors to the idea of LLM-based neural theorem proving. Specifically, two predominant paradigms have been extensively explored in neural

---

[*] These authors contributed equally. [†] Corresponding authors
[1] Code available at https://github.com/wiio12/LEGO-Prover.

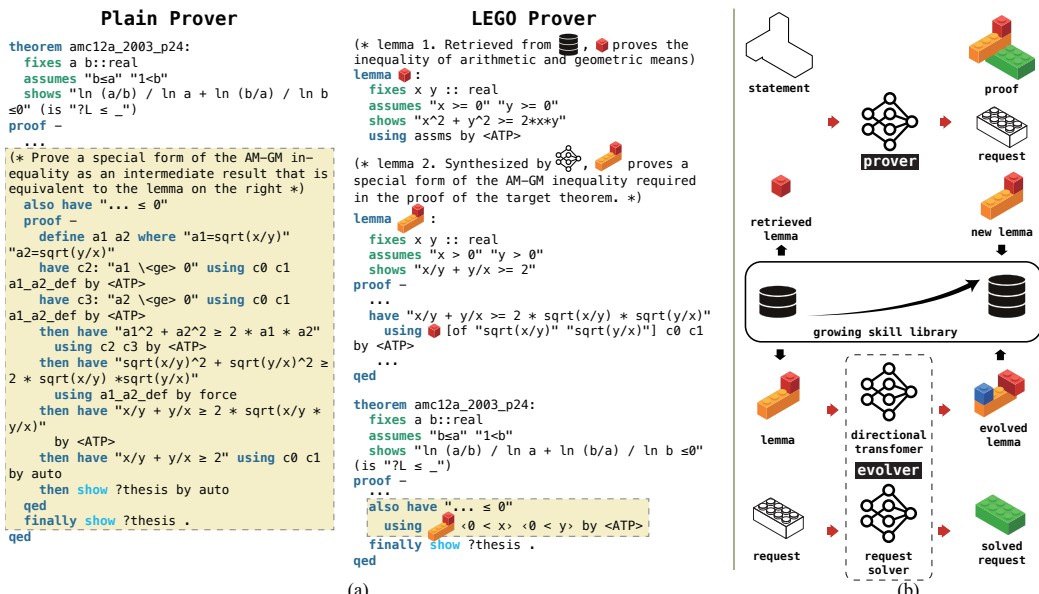

Figure 1: **(a) Comparison between plain provers and LEGO-Prover.** Unlike plain provers that prove the theorem sequentially, LEGO-Prover constructs the proof in a modular paradigm. Useful lemmas can be directly retrieved from the skill library and used as part of the proof. A newly constructed lemma can also be added to the skill library to help the proof of other theorems. **(b) Overall framework of our proposed LEGO-Prover.** The LEGO-Prover comprises two main components: the prover and the evolver. The prover modularly proves theorems using retrieved lemmas and, in the process, generates new lemmas and lemma requests as by-products. Meanwhile, the evolver either transforms lemmas for enhanced reusability and generalizability or directly addresses the requests. These components are interconnected through a growing library, which archives both verified lemmas and unsolved requests.

theorem proving. One stream of work involves step-by-step proof generation (Polu & Sutskever, 2020a; Han et al., 2022; Polu et al., 2022; Lample et al., 2022; Wang et al., 2023b; Yang et al., 2023; Jiang et al., 2022a), where fine-tuned models provide single-step proof actions coupled with search algorithms to find the complete proofs. Another paradigm leverages the coding capabilities of LLM to construct entire proofs in a single decoding process (Jiang et al., 2022b; Zhao et al., 2023; First et al., 2023). As shown in Fig. 1(a) left, these approaches share common proving strategies that synthesize the proof sequentially, with each step building upon the previous proof step, and stocking all the proof code into one large proof block. We denoted these approaches as *plain provers* since they generate the whole proof directly. Despite their promising results, plain provers still have several shortcomings. On one hand, plain provers attempt to prove theorems using static LLMs independently, while different problems can usually provide some insights into others. In other words, different problems may share the same lemmas while plain provers cannot utilize the proved lemmas once again even though it has been proved. On the other hand, even though plain provers can generate short-chain proofs with the help of advanced LLMs like ChatGPT or GPT-4 (OpenAI, 2023) , it usually fails when it comes to long-chain proofs due to the reasoning difficulty.

Inspired by the modularity of LEGO building blocks, we present *LEGO-Prover*, a novel approach designed to prove theorems in a block-by-block manner backed by a growing skill library. As shown in Fig. 1(a) right, LEGO-Prover tackles the problem of proving a theorem by first proving the sub-goal lemmas and then finalizing the problem using these lemmas. These lemmas can be retrieved from the skill library or newly constructed during the proving process. Specifically, Fig. 1(b) shows the overall process of LEGO-Prover, containing a prover and an evolver that share an expanding skill library. The prover takes the problem's formal statement as input and retrieves skills to prompt the LLM in generating the modular proof, with the generated lemmas accumulated into the skill library. However, lemmas created by the prover are often problem-specific with low reusability. Thus, LEGO-Prover incorporates an evolver that transforms the skills in the library for better generality, reusability, and complexity of the skills. The evolved new skills will also be verified and added back to the skill library. LEGO-Prover runs the prover and evolver in parallel (Algorithm 1). For each problem in the dataset, the prover makes 100 attempts, while the evolver continuously works to evolve new skills or solve requests, providing more lemmas for the prover.

We conduct extensive experiments on the popular theorem-proving dataset miniF2F (Zheng et al., 2021). LEGO-Prover significantly outperforms previous approaches, achieving a pass rate of 57.0% and 50.0% on the miniF2F valid and test datasets, respectively. With a 6.75% absolute improvement on average over the previous state-of-the-art methods. Our case study reveals that LLMs prove theorems modularly akin to LEGO block assembly, utilizing the retrieved skill by directly copying or using as a referee to construct the proof. Moreover, the learned skill library contains more than 20,000 skills encompassing many useful lemmas, as is shown in our case study and ablation study.

## 2 RELATED WORKS

**Machine learning for formal mathematics.** Modern formal mathematics environments often center around Interactive Theorem Provers (ITPs) like Lean (de Moura et al., 2015), Isabelle (Paulson, 1994), and Coq (Barras et al., 1997). These ITPs often include specific formal languages, accompanied formal verifiers, and automated provers like Sledgehammer. ITPs provide machine-human interactive interfaces, which give verification results during formal proof construction for specific theorems, and human coders can correct errors or continue to fill gaps in proofs under the guidance of error messages and local proof states, respectively.

**Proof search and premise selection.** Research leveraging machine learning techniques atop these formal mathematics environments generally falls into two predominant paradigms. The first focuses on proof search strategies and premise selection, epitomized by GPT-f (Polu & Sutskever, 2020a), where a language model advises single-step actions based on the current proving state, and the tree search finds a sequence of correct steps using actions given by the language model. The follow-up works PACT (Han et al., 2022) and Expert Iteration (Polu et al., 2022) incorporate supplemental pre-training tasks like theorem naming to enhance the policy model's reasoning ability. HTPS (Lample et al., 2022) applies Monte-Carlo tree search coupled with online training to optimize the exploration of the proof space. DT-Solver (Wang et al., 2023b) enhances search efficiency by dynamically adjusting search budgets to accommodate varying levels of state complexity. Thor (Jiang et al., 2022a) blends traditional Automated Theorem Provers (ATPs) with neural policy models to prove theorems in a neural-symbolic manner. Magnushammer (Mikuła et al., 2023) augments Thor's performance by integrating premise selection, thereby boosting the performance of rule-based ATPs.

**Autoformalization.** The second paradigm in machine learning for formal mathematics leverages the capabilities of LLMs for the formalization of mathematical proofs. (Wu et al., 2022) makes its first attempt at employing an LLM to translate natural language mathematical problems into formal theorem statements. Building on this, DSP (Jiang et al., 2022b) uses natural language as guidance to fully formalize proofs. (First et al., 2023) goes a step further by generating complete proofs in a single pass and introducing a proof repair model to enhance the theorem-proving capabilities. (Zhao et al., 2023) advances DSP by incorporating cross-verified informal proofs to better inform the generation of formal proof sketches. Despite their contributions, none of the aforementioned methods have succeeded in establishing a learning paradigm that incrementally formalizes increasingly complex problems via a growing skill library, a gap that our work seeks to fill.

**Skill-based agents.** LEGO-Prover is also related to trending AI agents powered by LLMs (Shen et al., 2023; Park et al., 2023). Voyager (Wang et al., 2023a) creates an AI agent capable of autonomously playing Minecraft. It has a dynamic growing skill library that empowers the in-game character to tackle increasingly intricate tasks. Similarly, (Cai et al., 2023) showcases the ability to generate reusable Python tools and documentation. DreamCoder (Ellis et al., 2020) develops problem-solving expertise by creating and expanding programming languages, guided by a neural network and a 'wake-sleep' learning algorithm. Lastly, template-based conjecturing (Nagashima et al., 2023) introduces a novel approach to generate auxiliary lemmas and use them to prove final goals.

## 3 METHOD

In this section, we introduce the detailed implementations of our proposed LEGO-Prover. Following the setting of (Jiang et al., 2022b), we assume that each theorem is equipped with an informal statement, a human-written informal proof, and a formal statement defining the problem. In the subsequent sections, we will provide detailed introductions to the skill library, the prover, and the evolver. A detailed algorithm description is provided in Appendix A.2.

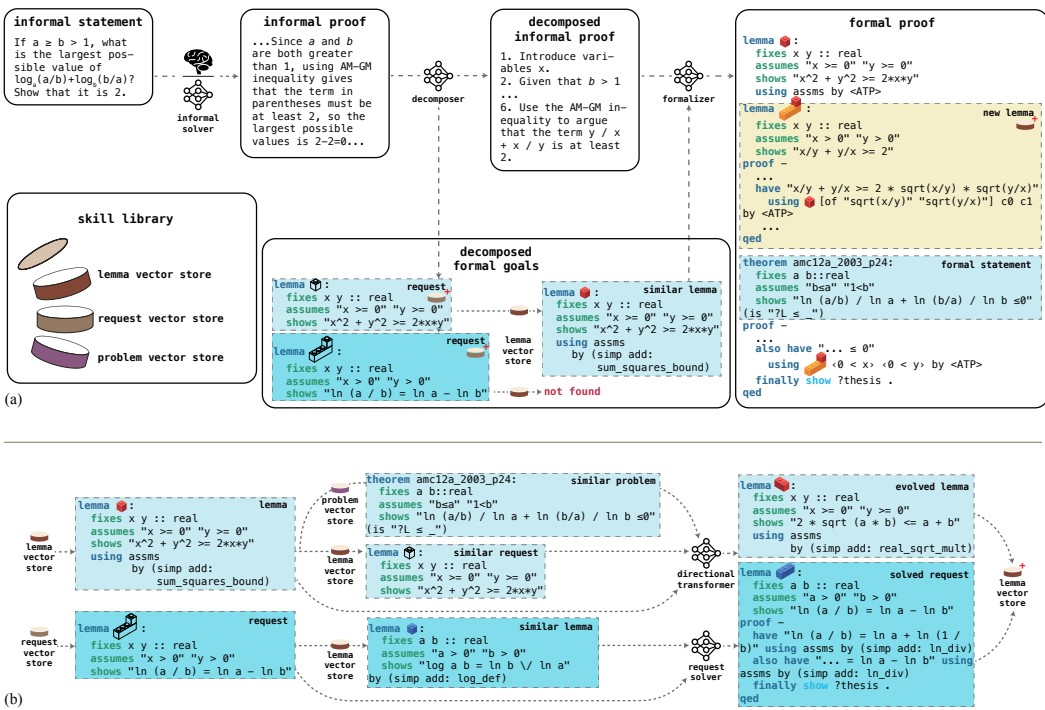

Figure 2: **Overview of LEGO-Prover.** (a) The prover follows three steps to solve a problem: first, the informal solver generates a solution to the problem statement. This solution is then broken down by the decomposer into a step-by-step proof, and suggesting useful sub-goals as lemma statements. We use these statements to retrieve useful lemma from the skill library. Finally, the formalizer generates the final proof using the decomposed informal proof and the retrieved lemmas in a block-by-block fashion. LEGO-Prover subsequently adds the proposed requests and newly constructed lemmas to the request store and lemma store, respectively. (b) The evolver contains two evolution approaches, the directional transformer transforms the existing skill into a more general and reusable form in four predefined directions. The request solver directly solves the requested subgoal proposed by the prover. Formally verified lemmas from the evolver are added to the lemma vector store.

## 3.1 SKILL LIBRARY

The skill library contains various vector stores that are optimized for retrieval. Every vector store maintains its data in pairs consisting of documents and their corresponding embeddings, encoded with an embedding language model[2]. Upon the receipt of a query, the vector store employs the embedding model to encode the query and leverages the k-NN algorithm to retrieve relevant documents stored within. The skill library used in LEGO-Prover is comprised of three distinct vector stores. 1) The lemma vector store contains the Isabelle verified lemmas, encompassing both the lemma's statement and its proof. This is the core of the skill library and it facilitates the perpetual enhancement of LLMs' capabilities in constructing increasingly intricate theorems. For representation simplicity, the notion of lemmas in the lemma vector store and skills in the skill library are use interchangeably in this paper. 2) The request vector stores preserve the lemma statements proposed by the decomposer. These requests are crucial to the success of LEGO-Prover, their works as an in-depth reasoned query for retrieving the useful skill for the prover, as well as possible complete lemmas when they are solved by the evolver. 3) The problem vector store houses the formal statements in the miniF2F dataset. The evolver utilizes these problem statements as heuristics to guide the LLMs in generating more beneficial new lemmas.

## 3.2 PROVER

As illustrated in Fig. 2 (a) and Algorithm 2, the prover employs a three-step process to construct the proof. Initially, a LLM is deployed as an informal problem solver, drafting solutions in natural language that correspond to the informal statements presented. Similar to the approach discussed in

---

[2] Practically, ChromaDB serves as our vector store, coupled with the OpenAI's text-davinci-ada embedding model.

Table 1: Prompt outline for four components in LEGO-Prover.

| Decomposer | Formalizer | Directional transformer | Request solver |
|---|---|---|---|
| *Input:*
... provide a better structured step-by-step proof that is closer to Isabelle. and request relevant lemmas, and theorems that might help in proving this problem.
{in-context example}...
**Statement:** {informal statement}
**Informal proof:** {informal proof}
**Formal statement:** {formal statement}
*Output:*
**Structural proof:** step 1...
**Required skills:**
**Thought 1:** {CoT for request 1}
**Code 1:** {lemma statement}
... | *Input:*
... provide formal proof in response to a given problem statement...
**Useful skills 1:** {lemma code}
**Useful skills 2:** {lemma code}
...
{in-context example}...
**Statement:** {informal statement}
**Informal proof:** {structural informal proof}
**Formal statement:** {formal statement}
*Output:*
**Foraml proof:** complete formal proof code | *Input:*
...your task is to modify the given lemma, theorem, function, or definition given in the code to aid in solving one or more of the problems provided. You should accomplish this by {transform direction describtion}...
**Problem 1:** {problem/request}
**Problem 2:** {problem/request}
...
{in-context example}...
**Skill to evolve:** {lemma code}
*Output:*
**Evolved skill:** {new lemma code} | *Input:*
...provide a formal proof in response to a given formal statement...
{retrived lemma as in-context example}...
**Formal statement:** {problem statement}
*Output:*
**Formal proof** {new lemma code} |

(Jiang et al., 2022b), LEGO-Prover explores the use of both ground truth, which are human-written proofs, and model-generated proofs as viable alternatives. After obtaining the informal proof, LEGO-Prover methodically builds the formal proof. This process involves two sequential stages: first, the *decomposer* breaks down the proof into manageable components; second, the *formalizer* rigorously structures these components into a formal proof.

**Decomposer.** The decomposer aims to decompose the formalization tasks, which transform the informal proof into the decomposed step-by-step informal proof as well as decompose the problem into formal goals. As shown in Table. 1 column 1 (and Fig. 6 for complete example), the decomposer prompts the LLM to refine informal proofs, producing step-by-step informal proof that more closely aligns with the structure of the actual Isabelle proof code. We posit that this alignment is crucial as it considerably reduces the complexity encountered during the formalization process. Concurrently, the decomposer tasks the LLM with generating requests: some potential lemma or theorem statements that could be useful in addressing the given problem. Each request is composed of a chain-of-thought reasoning on what kind of lemma is required for solving the problem followed by the formal statement of the lemma. Subsequently, LEGO-Prover put these requests into the request vector store.

**Formalizer.** As depicted in Table. 1 column 2 (full example shown in Fig. 7), the process of formalization involves translating an informal proof into Isabelle's proof sketches. The formalizer employs the proposed request originating from the decomposer and the formal statement of the problem as query texts and, in total, retrieves $n_f$ skills. Upon collecting all the necessary input, the LLM is tasked to provide the proof code. Unlike the setting in (Jiang et al., 2022b) and (Zhao et al., 2023), we prompt the LLM to construct the complete content of the source file in Isabelle. This may encompass the requisite imports, definitions, or lemmas before the elucidation of the main problem needs to be proven. Consequently, the LLM possesses the capability to generate or apply useful lemmas before embarking on the resolution of the problem. Empirical evaluations demonstrate that our model exhibits a more modular problem-solving approach compared to established baseline methods. This modularity facilitates recycling smaller lemma components, thereby enhancing the LLM's capability to tackle progressively intricate problems.

After obtaining the formalized proof code, LEGO-Prover employs the Isabelle theorem prover to verify the correctness of the provided proof code. In instances where a particular proof tactic (such as "by ...") falls short of proving the given conjecture, we resort to 11 heuristic tactics alongside the sledgehammer method to facilitate an auto-correction. The heuristic selection we employ is consistent with those presented in (Jiang et al., 2022b). After verifying the code, all validated lemmas or theorems are added to the skill vector store, while any failed lemmas' statement is added to the request vector store.

### 3.3 EVOLVER

The lemmas extracted from the prover are mostly problem-specific, rendering them non-reusable with limited applicability. And the number of these lemmas is also very limited. The objective of the evolver is to create or refine these skills, enhancing their reusability and expanding their functional coverage. As shown in Fig. 2 (b), the evolver contains two functionalities: the directional transformer transforms the current skill and the request solver directly solves the request proposed by the prover to create new lemmas. Algorithm 3 also illustrates the overall process for the evolver.

**Directional transformer.** As shown in Table 1 column 3 (full example in Fig. 9), the objective of the directional transformer is to facilitate the evolution of a skill along various predefined trajectories, thereby augmenting the reusability and usefulness of the skill. It is composed of four distinct trajectories: extension of dimensions, identification of key concepts, parameterization, and enhancement of complexity. Table B shows the detailed functionality of each evolving direction. Each instance of the directional transformer adheres to a unified prompt template depicted in Fig. 9. The adaptation involves substituting the core description and its in-context examples for the corresponding evolving directions. Specifically, the directional transformer begins with randomly selecting the least evolved skill (with the least amount of time being selected to evolve). Subsequently, the directional transformer employs this skill to retrieve $n_d$ relevant pending problem's formal statement from the problem vector store and the relevant request's formal statement from the request vector store. Upon assembling the inputs for the LLM, the directional transformer arbitrarily selects one direction of evolution and prompts the LLM to generate a novel skill.

**Request solver.** The request solver is designed to facilitate the development of new skills by directly addressing the sub-goals proposed by the prover. As shown in Table 1 column 4 (full example in Fig. 8, the process initiated by the request solver involves the random selection of a minimally solved request (with least amount of time being selected to solve the request). After this selection, this request is employed to query the lemma vector store to retrieve pertinent skills that serve as in-context demonstration examples. Finally, the request solver prompts the LLM to generate the proof for the request. It should be noted that these requests, which are conjectures proposed by LLMs without verified proof, could potentially be incorrect. Only those conjectures that can be substantiated with a valid proof are accepted by a formal verifier. The challenge lies in the fact that determining the satisfiability of a conjecture, especially within complex mathematical theories, is an undecidable problem. Consequently, no purely symbolic method can guarantee the rejection of these incorrect conjectures. While conjectures with illegal syntax are straightforwardly rejected by the verifier, those that are semantically incorrect can only be incrementally dismissed after multiple unsuccessful attempts to prove them using LLMs.

After obtaining the new skill (evolved lemma or solved request) generated by the LLM, the evolver uses Isabelle to verify the generated code. To reduce redundancy in the skill library, we compare new skills against existing ones using the `SequenceMatcher` method from Python's `difflib`. Only skills that are verified and show a difference below the threshold of 0.85 are added to the library.

## 4 EXPERIMENTS

### 4.1 EXPERIMENTAL SETUP

In this section, we introduce the experiment setup for LEGO-Prover. Consistent with the (Jiang et al., 2022b; Zhao et al., 2023), each problem undergoes 100 attempts of proving using ChatGPT (GPT-3.5). More implementation details are provided in Appendix A.1.

**Dataset and evaluation.** For a more accurate comparison, we follow (Jiang et al., 2022b; Zhao et al., 2023) and adopt the miniF2F dataset (Zheng et al., 2021). This dataset includes 488 problems that vary in difficulty, ranging from basic algebra and number theory, originating from the MATH dataset (Hendrycks et al., 2021), to more challenging problems found in the AIME and IMO. The problems are divided into valid and test sets, with 244 problems each[3]. LEGO-Prover employs the updated miniF2F dataset from (Jiang et al., 2022b), featuring questions with formal and informal statements, along with human-written informal proofs.

**Baseline methods.** We have included baselines that represent state-of-the-art neural theorem proving in Isabelle. Thor(Jiang et al., 2022a) and Thor with expert iteration on auto-formalized data (Wu et al., 2022) are works focused on proof search paradigms, which use a fine-tuned 700m language model to prove theorems. Draft, Sketch, and Prove (Jiang et al., 2022b) and Subgoal-Learning (Zhao et al., 2023) are works that use Codex or ChatGPT to prove theorems directly.

Following the setting from (Jiang et al., 2022b), we test the LEGO-Prover with model-generated and human-written informal proofs. The model-generated informal proofs are pre-generated using

---

[3]LEGO-Prover does not require training, thus these two splits are treated identically, displaying separate results only for detailed comparisons with prior methods.

Table 2: **Proving success rates on the miniF2F dataset with Isabelle.** The table displays the success rates of previous works and the LEGO-Prover, The highest success rates for each set are highlighted in bold. LEGO-Prover* denotes the cumulative pass rate of the miniF2F dataset, considering the total number of problems solved using model-generated and human-written informal proofs.

| Success rate | LLM | miniF2F-valid | miniF2F-test |
|---|---|---|---|
| *Baselines* | | | |
| Thor (Jiang et al., 2022a) | - | 28.3% | 29.9% |
| Thor + expert iteration (Wu et al., 2022) | Codex | 37.3% | 35.2% |
| Draft, sketch, and Prove (Jiang et al., 2022b) | Codex | 42.6% | 39.3% |
| Subgoal-Learning (Zhao et al., 2023) | ChatGPT | 48.0% | 45.5% |
| *Ours (100 attempts)* | | | |
| LEGO-Prover (model informal proof) | ChatGPT | 52.0% | 45.5% |
| LEGO-Prover (human informal proof) | ChatGPT | 55.3% | **50.0**% |
| LEGO-Prover* | ChatGPT | **57.0**% | **50.0**% |
| *Ablations (100 attempts)* | | | |
| - Skill Library (human informal proof) | ChatGPT | 50.4%($-4.9$%) | - |

GPT-4, with up to 20 informal proofs per problem (12.13 on average). For each proving attempt, we randomly select one proof as the informal proof to feed into the decomposer procedure. For the ablation study, we remove the growing skill library to validate the effectiveness of the LEGO-Prover using human-written informal proofs. In this setup, the prover operates as usual, but we ignore the requests provided by the decomposer and supply an empty list of reference skills to the formalizer. As a result, the evolver is not utilized. Due to limited resources and the expense of OpenAI API calls[4], we perform ablation only on the miniF2F validation set.

## 4.2 MAIN RESULT

In Table. 2, we illustrate the proportion of successful formal proofs found on the miniF2F dataset. Our proposed LEGO-Prover significantly outperforms both search-based and LLM-based methods. With proofs written by humans, the LEGO-Prover improves by 7.3% and 4.5% over the Subgoal-Learning method on the miniF2F-valid and miniF2F-test datasets, respectively. A total of 257 out of 488 problems were solved by the LEGO-Prover with human-written proof. When replacing human-written informal proofs with model-generated informal proofs, the LEGO-Prover still achieves 52.4% and 45.5% pass rates on the valid set and test set, respectively, close to the results with human-written informal proofs.

**Effects of the growing skill library.** The growing skill library greatly enhances the proving ability of static LLMs like ChatGPT or GPT-4. As the major contribution of LEGO-Prover, we are interested in how well it contributes to solving more problems and improving the LLMs' ability. Specifically, we remove the growing skill library (including the evolver). As shown in the Table. 2, LEGO-Prover achieves 55.3% on the validation set, whereas the LEGO-Prover without a skill library achieves 50.4%. For a more intuitive representation, we further plot the trends of the number of problems solved against the number of proving attempts for both settings, shown in Fig. 3(a). Compared to the problem solved without a growing skill library, the advantage of adding the skill library is initially minimal, as the libraries are still underdeveloped and lack useful skills. However, as the skill library expands, the gap between LEGO-Prover and the ablation method widens consistently (the performance gap in 50 attempts is 3.3% which increases to 4.9% in 100 attempts). This outcome supports our hypothesis that the prover becomes increasingly adept at formalizing theorems as more skills are added to the skill library.

## 4.3 ANALYSIS

### 4.3.1 WHAT DOES THE GROWING SKILL LIBRARY LOOK LIKE?

Figure. 3(c) illustrates an example of a skill-evolving tree in the skill library. The grown skill library is a massive forest containing numerous evolved trees like this one. The lemmas, originating from either the prover or the evolver's request solver sub-task (as the example shown in the figure), become

---

[4] Estimated to be around 600 dollars for one experiment with 100 proof attempts

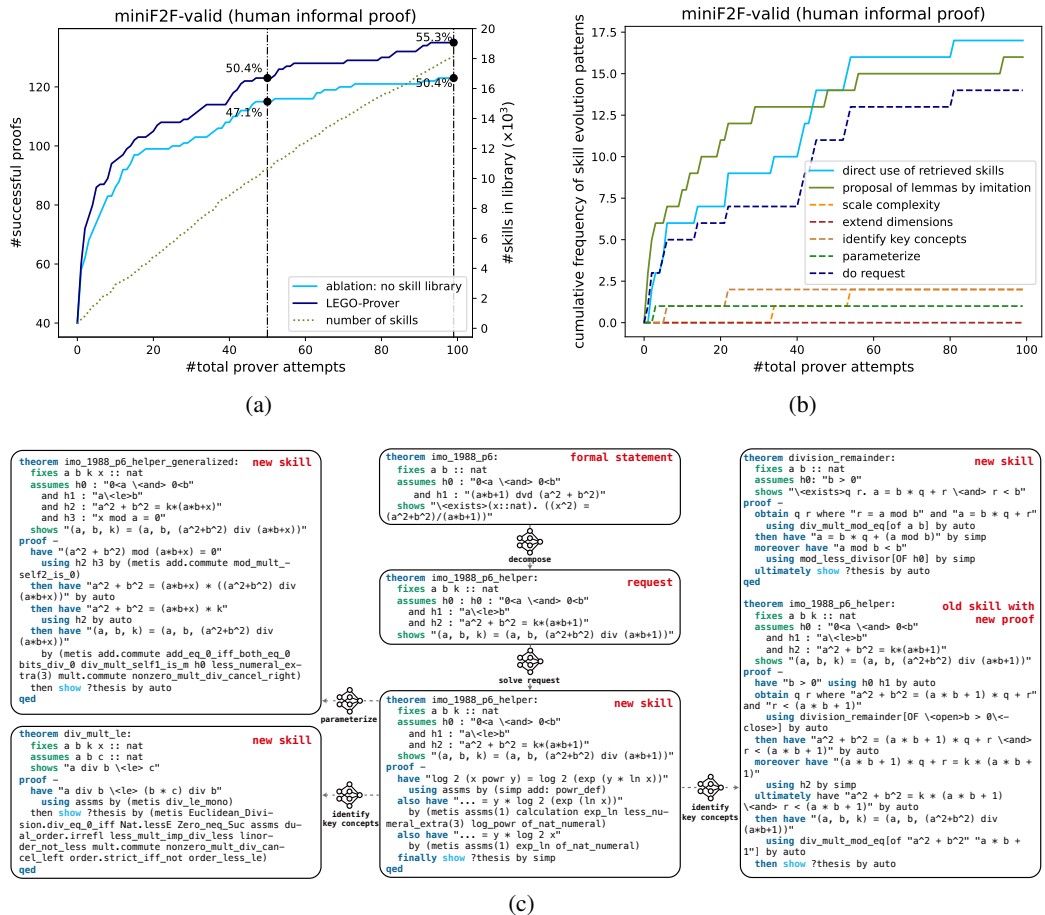

Figure 3: (a) A comparison of proof success rate between LEGO-Prover with and without the growing skill library. (b) Distribution of skill origins in successful proofs, shown against prover attempts and their percentage contributions to total successes. Solid lines represent the distribution of skill usage methods, while dotted lines detail the specific skill origins. (c) A skill-evolving tree gradually generated through multiple steps for `imo_1988_p6` conceals how a relatively large-scale skill library is produced from a few seed theorems.

the root nodes of these skill-evolving trees. The evolver's directional transformation generalizes these lemmas and creates child nodes. In terms of statistics, there are 22532 skills in the skill library in total, with 10.8% of the skills originating from the prover, 38.2% originating from the evolver's solve request sub-task, and 51.1% originating from the evolver's directional transformation sub-tasks. Although some lemmas generated by LEGO-Prover are trivial or already in Isabelle's theorem library, most are unique, interesting, and useful. The main challenge in auto-formalization lies in bridging the gap between natural and formal mathematical languages; a simple step in natural language often translates to numerous lines in formal language. However, as more lemmas and theorems are accumulated, this gap narrows. Ultimately, LEGO-Prover's expanded skill library significantly contributes to bridging this divide.

### 4.3.2 How does the skill boost the ability of LLMs?

To investigate closely how these learned skills can better help and boost the performance of LLM, we manually inspect the successfully proven problems in the miniF2F valid set. The conclusions are as follows:

**Skill as directly reusable blocks.** This is the most straightforward way of utilizing the generated skills. Since every skill in the skill library is verified by the Isabelle prover, LLM can directly copy the lemma code from the input prompt without fear of causing an error. As shown in Fig. 4 left, the final proof of the problem `algebra_amgm_faxinrrp2msqrt2geq2mxm1div2x` directly copies the retrieved skill `am_gm`'s code as part of the proof and uses this lemma to help prove the problem.

```
                    retrieved skill
lemma am_gm:
  fixes x :: real
  assumes "x > 0"
  shows "x + 1/(2*x) ≥ sqrt 2"
proof -
  have "(sqrt x - sqrt (1/(2*x)))^2 ≥ 0" by simp
  then have "x + 1/(2*x) - 2 * sqrt x * sqrt (1/(2*x)) ≥ 0"
    by (smt (verit) add_le_imp_le_diff assms divide_nonneg_nonneg less_eq_-
  real_def real_sqrt_pow2 sum_squares_bound)
  then have "x + 1/(2*x) - sqrt 2 ≥ 0"
    by (metis add_divide_distrib assms div_by_1 divide_divide_eq_right di-
  vide_self less_numeral_extra(3) mult_2 one_add_one real_div_sqrt re-
  al_sqrt_divide real_sqrt_eq_1_iff real_sqrt_gt_0_iff real_sqrt_one times_di-
  vide_eq_right zero_le_numeral)
  then have "x + 1/(2*x) ≥ sqrt 2" by simp
  then show ?thesis by simp
qed
```

```
                    synthesized proof
lemma am_gm:
  fixes x :: real
  assumes "x > 0"
  shows "x + 1/(2*x) ≥ sqrt 2"
proof -
  have "(sqrt x - sqrt (1/(2*x)))^2 ≥ 0" by simp
  then have "x + 1/(2*x) - 2 * sqrt x * sqrt (1/(2*x)) ≥ 0"
    by (smt (verit) add_le_imp_le_diff assms divide_nonneg_nonneg less_eq_-
  real_def real_sqrt_pow2 sum_squares_bound)
  then have "x + 1/(2*x) - sqrt 2 ≥ 0"
    by (metis add_divide_distrib assms div_by_1 divide_divide_eq_right di-
  vide_self less_numeral_extra(3) mult_2 one_add_one real_div_sqrt re-
  al_sqrt_divide real_sqrt_eq_1_iff real_sqrt_gt_0_iff real_sqrt_one times_di-
  vide_eq_right zero_le_numeral)
  then have "x + 1/(2*x) ≥ sqrt 2" by simp
  then show ?thesis by simp
qed

theorem algebra_amgm_faxinrrp2msqrt2geq2mxm1div2x:
  "⋀x. (x>0) ⟹ 2 - sqrt 2 ≥ 2 - x - 1/ (2 * x)"
proof -
  fixes x :: real
  assumes h0: "x>0"
  (*Step 1: We want to show that $x + \frac{1}{2x} \geq \sqrt{2}$ is true.*)
  (* Step 2: Apply the AM-GM inequality to $x$ and $\frac{1}{2x}$ to get $x
  + \frac{1}{2x} \geq 2\sqrt{\frac{x}{2x}}$. *)
  (* Step 3: Simplify $2\sqrt{\frac{x}{2x}}$ to $\sqrt{2}$. *)
  (* Step 4: Conclude that $x + \frac{1}{2x} \geq \sqrt{2}$ is true. *)
  have c1: "f (500 * (6/5)) = f 500 / (6/5)"
    have "x + 1/(2*x) ≥ sqrt 2" using am_gm[OF h0] by simp
  (* Step 5: Since $2 - \sqrt{2} \geq 2 - x - \frac{1}{2x}$ is equivalent to
  $x + \frac{1}{2x} \geq \sqrt{2}$, we can conclude that $2 - \sqrt{2} \geq 2
  - x - \frac{1}{2x}$ is true. *)
  then show "2 - sqrt 2 ≥ 2 - x - 1/ (2 * x)" by simp
qed
```

(a) Directly Use

```
                    retrieved skill
lemma prod_1n_4n:
  fixes n :: nat
  shows "(∏k = 1..n. (4 * k)) = 4^n * fact n"
proof(induct n)
  case 0
  then show ?case by simp
next
  case (Suc n)
  then show ?case by (simp add: algebra_simps)
qed
```

```
                    synthesized proof
lemma prod_frac_common_factor:
  fixes n:: nat and a::real
  assumes "a ≠ 0"
  shows "(∏k = 1..n. (a * k + a) / (a * k)) = n + 1"
proof(induct n)
  case 0
  then show ?case by simp
next
  case (Suc n)
  have "(∏k = 1..Suc n. (a * k + a) / (a * k)) = ((a * Suc n + a) / (a * Suc
  n)) * (∏k = 1..n. (a * k + a) / (a * k))"
    by simp
  also have "... = ((a * Suc n + a) / (a * Suc n)) * (n + 1)"
    using Suc by simp
  also have "... = Suc n + 1"
    using assms by (smt (verit) add.commute add_divide_distrib divisors_zero
  nonzero_mult_div_cancel_left nonzero_mult_divide_mult_cancel_right of_nat_1
  of_nat_add of_nat_neq_0 plus_1_eq_Suc right_inverse_eq times_divide_eq_left)
  finally show ?case by simp
qed

theorem amc12a_2008_p4:
  "(∏k::nat=1..501. ((4::real) * k + 4) / (4 * k)) = 502"
proof -
  (* Step 1: Rewrite the given product as $\frac {8}{4}\cdot\frac {12}{8}\c-
  dot\frac {16}{12}\cdots\frac {4n + 4}{4n}\cdots\frac {2008}{2004}$. *)
  have "(∏k::nat=1..501. ((4::real) * k + 4) / (4 * k)) = (∏k::nat=1..501.
  (4 * (k + 1)) / (4 * k))"
    by eval
  (* Step 2: Simplify the product by canceling out common factors. Notice
  that each term in the numerator cancels with the corresponding term in the
  denominator, leaving only the last term $\frac {2008}{4}$. *)
  also have "... = (∏k::nat=1..501. (k + 1) / k)"
    by eval
  (* Use lemma 1 to simplify the product *)
  also have "... = 501 + 1"
    using prod_frac_common_factor[of "1::real" "501"] by eval
  (* Step 3: Calculate the value of $\frac {2008}{4}$ to find that it is
  equal to $502$. *)
  also have "... = 502"
    by simp
  (* Step 4: Conclude that the given product is equal to $502$. *)
  finally show ?thesis by simp
qed
```

(b) Propose Lemma by Imitation

Figure 4: Two primary forms of utilizing the skills: **directly use** or **propose lemma by imitation**

**Skill as reference for solving the problem.** Many skills cannot be directly reused but are very helpful as reference examples for formalizing the main problem. As shown in Fig. 4 right, the retrieved skill examples `prod_1n_4n` provide great clues for solving the conjecture `prod_frac_common_factor`. Since the provided skills are lemmas with verified steps, these steps drastically increase the accuracy of the LLM to generate the correct proof steps.

Fig. 3(b) first compares two scenarios: directly applying retrieved skills to the proofs and constructing new lemmas by imitating retrieved skills to assist in theorem proving (shown in the solid line). We examine manually to determine the skill evolution pattern of the lemmas used in the proofs (shown in the dotted line). Out of 135 problems of the miniF2F-valid dataset passing the validation of the Isabelle verifier, 24% is completed with the aid of retrieved skills. Within this subset, 51% of the problems directly incorporate the retrieved skills into their proofs, while the remaining 49% formulate new lemmas that are specifically tailored to address the distinct problems at hand. Regarding the skills directly applied in the proofs, 71% are procured by the "do requests" procedure. The skills derived through the evolution techniques of "identifying key concepts" and "scaling complexity" each contributes to 12%, while those acquired through "parameterization" constitute 6%. Although skill as directly reusable blocks is the most ideal usage of skills in the library, the problems solved by directly reusing the skill are not substantial. That is because many trivial problems in the dataset miniF2F can be solved trivially without requiring any skill as a reference.

## 5 CONCLUSIONS

In this work, we introduced a new theorem-proving method, LEGO-Prover, which uses a growing skill library to continuously boost the capability of LLM for formalization. The prover utilizes refined structural informal proof and retrieved lemma to correctly formalize the proof. The evolver solves the request proposed by the prover or evolves existing skills into new skills. LEGO-Prover introduces a fundamentally different theorem proving paradigms for the community. With the previous approaches all struggling to complete the proof at once, LEGO-Prover tries to prove the theorem in a block-by-block manner, akin to divide and concur approaches. Extensive tests show that our method can indeed improve pass rates on the miniF2F dataset. Our ablation studies and detailed analysis showcase the effectiveness of each component we proposed in LEGO-Prover.

## ACKNOWLEDGEMENTS

This work was supported in part by the National Key R&D Program of China under Grant No. 2020AAA0109700, Guangdong Outstanding Youth Fund (Grant No. 2021B1515020061), National Natural Science Foundation of China (NSFC) under Grant No.61976233, Mobility Grant Award under Grant No. M-0461, Shenzhen Science and Technology Program (Grant No. GJHZ20220913142600001), Nansha Key RD Program under Grant No.2022ZD014, National Natural Science Foundation of China (U2001211, U22B2060), Research Foundation of Science and Technology Plan Project of Guangzhou City (2023B01J0001, 2024B01W0004), and National Natural Science Foundation of China under Grant No.62006255. We thank MindSpore for the partial support of this work, which is a new deep learning computing framework[5].

Xiaodan Liang is with the School of Intelligent Systems Engineering, Sun Yat-sen University at Shenzhen, Shenzhen 518107, China, and also with DarkMatter AI Research Guangzhou 511458, China.

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

## A  MORE DETAILS ON LEGO-PROVER

### A.1  IMPLEMENTATION DETAILS

To expedite the experimental procedure, both the prover and the evolver are executed in a multi-processing manner, maintaining a process number ratio of $3 : 8$ respectively. In order to maximally leverage the expanding skill library, problems are formalized through successive rounds, with each round addressing each valid/test set problem once.

For the execution of the prover and the evolver, ChatGPT is utilized as the LLM.[6] The temperature is consistently set at $T = 0.7$ across all procedures. Within the prover, 3-shot examples are leveraged for the decomposer. Regarding the formalizer, the quantity of reference skills $n_f$ is set to 6 for the valid set and 4 for the test set, and paired with 2 formalization in-context examples. For the directional transformer, the number of reference problem statements $n_d$ is set to 4, supplemented by two directional transformation in-context examples. For the request solver, 3 skills are retrieved for forming the in-context demonstration examples.

For interacting with the Isabelle theorem prover, we employ the PISA environment (Jiang et al., 2021). PISA is a flexible Python REPL wrapper for Isabelle, capable of verifying Isabelle's code and providing information such as proof states or error messages from Isabelle. We consider a formalized proof valid if and only if (a) the proof does not contain "cheating" keywords (`sorry` or `oops`) that exit a proof without completing it. (b) Isabelle can verify the proof code containing the corresponding formal statement.

### A.2  ALGORITHM DESCRIPTION

In this section, the pseudo-code for our proposed LEGO-Prover is presented. Initially, the main process of LEGO-Prover is introduced in Algorithm 1, followed by detailed descriptions of the prover process in Algorithm 2, and the evolver process in Algorithm 3.

**LEGO Prover main process.** Algorithm 1 details the parallel execution of provers and evolvers in the main process. The algorithm begins by taking the miniF2F dataset as input, which comprises 244 problems per split, each featuring an informal statement, an informal proof, and a formal statement. Lines 2-5 initialize three vector stores using `ChromaDB`, a vector database. The `mp.Queue` introduced in line 6 is a Python multiprocessing queue, facilitating synchronization of queue elements across different processes. Each problem is replicated `n_attempts_per_problem` times and added to the `miniF2FQueue`. Lines 8-13 launch `n_prover` prover processes and `n_evolver` evolver processes.

---

**Algorithm 1** LEGO-Prover main process

---

1: **Input:** miniF2FData (containing 244 problems, each with an informal statement, informal proof, and formal statement)
2: **Initialize:**
3: LemmaS ← ChromaDB()  //lemma vector store, initially empty
4: RequestS ← ChromaDB()  //request vector store, initially empty
5: ProblemS ← ChromaDB(miniF2FData)  //problem vector store, initialized with miniF2F problem formal statements.
6: miniF2FQueue ← mp.Queue(miniF2FData × n_attempts_per_problem)  //share queue
7: **Begin Parallel Execution**
8:   **For** $i = 1$ **to** $n\_prover$
9:     **Start Process:** prover(LemmaS, RequestS, miniF2FQueue)
10:   **End For**
11:   **For** $j = 1$ **to** $n\_evolver$
12:     **Start Process:** evolver(LemmaS, RequestS, ProblemS)
13:   **End For**
14: **End Parallel Execution**

---

[6] A combination of gpt-3.5-turbo, gpt-3.5-turbo-0301, gpt-3.5-turbo-0613, gpt-3.5-turbo-16k, and gpt-3.5-turbo-16k-0613 is employed, with a model being selected randomly during calls to the OpenAI API.

---

**Algorithm 2** Prover process

---

1: **Function** prover(LemmaS, RequestS, miniF2FQueue)
2: **while** miniF2FQueue not empty **do**
3:    `//infStmt, infProof, formStmt are abbreviations for informal`
      `statement, informal proof, and formal statement`
4:    infStmt, infProof, formStmt ← miniF2FQueue.pop()
5:    `//Use informal solver to produce model informal proof`
6:    **if** model informal proof required **then**
7:       infProof ← InformalSolver(infStmt)
8:    **end if**
9:    `//decomposer generate strucal step-by-step informal proof and lemma`
      `statements as requests.`
10:    strucInfProof, lemmaRequests ← Decomposer(infStmt, infProof, formStmt)
11:    `//add request to request vector store and retrieve relevant lemma`
      `from lemma vector store`
12:    RequestS.adds(lemmaRequests, init_update_count=0)
13:    retrievedLemmas ← LemmaS.retrieveKNN(lemmaRequests)
14:    `//formalizer generate complete proof code`
15:    proofCode ← Formalizer(infStmt, strucInfProof, formStmt, retrievedLemmas)
16:    `//Isabelle theorem prover verify the proof code and return`
      `proofResult (true/false), correct lemmas and failed lemma's`
      `statement as new request`
17:    proofResult, correctLemmas, newRequests ← IsabelleEnv.verify(proofCode)
18:    RequestS.adds(newRequests, init_update_count=0)
19:    LemmaS.adds(correctLemmas, init_update_count=0)
20: **end while**
21: **End Function**

---

**Prover process.** Algorithm 2 outlines the prover process. The prover employs the functions `Decomposer` (line 10) and `Formalizer` (line 15), which are LLM wrapper functions. These functions sequentially build the prompt text by filling the placeholders, query the LLM, and parse its output. Back to the algorithm, each instance of the prover process continually retrieves unsolved miniF2F problems from `miniF2FQueue` (line 2). Specifically, the prover extracts the informal statement, the human-written informal proof, and the formal statement from the problem data. In lines 6-8, the InformalSolver generates the informal proof if required[7]. In lines 10-13, the `Decomposer` is used to create a step-by-step structural informal proof and to generate lemma requests. These requests are first added to the request vector store and later utilized to retrieve corresponding lemmas from the lemma vector store. In lines 15-19, the `Formalizer` produces the proof code, which is then verified by the Isabelle theorem prover. Correct lemmas are added to the lemma vector store with their initial update counts set to zero. For lemmas that fail verification, their proof code is removed, and their statements are added to the request vector store.

**Evolver process.** Algorithm 3 describes the evolver process. The evolver runs parallelly with the prover and runs continuously until the prover completes all proving tasks. Similar to the `Decomposer` and `Formalizer`, both `DirectionalTransformer` and `RequestSolver` are LLM wrappers. Back to the algorithm, in each iteration, the evolver initially randomly selects an evolving type between `Directional Transformer` and `Request Solver` (line 3). Lines 4-14 describe the overall pipeline of directional transformer. Specifically, when `Directional Transformer` is chosen, the evolver randomly picks one of four transformation types. In lines 7-12, a lemma with the lowest update count is selected from the lemma vector store, and its update count is then incremented. The evolver uses this lemma to query the request and problem vector stores for relevant problems. In line 14, the `DirectionalTransformer` queries the LLM to transform the selected lemma into a new one. Lines 16-22 showcase the pipeline for the request solver. In lines 17-20, if the `Request Solver` is selected, the process begins by choosing the least updated request from the request vector store and incrementing its update count. This request is then used to query for relevant lemmas. The `RequestSolver` prompts the LLM to address the

---

[7]As described in Sec. 4.1, **Baseline method**, in practice, the pre-generated model informal proof is used.

request. In lines 25-27, the Isabelle theorem prover verifies the generated lemma code, and if correct, it is added to the lemma vector store.

---

**Algorithm 3** Evolver process

---

1: **Function** evolver(LemmaS, RequestS, ProblemS)
2: **while** TRUE **do**
3:     evolvingType ← Random.choice(['Directional Transformer', 'Request Solver'])
4:     **if** evolvingType == 'Directional Transformer' **then**
5:         transType ← Random.choice(['Identify key concepts', 'Parameterize', 'Scale complexity', 'Extend dimensions'])
6:         `//get a lemma in lemma vector store with least update counts.`
7:         selectedLemma ← LemmaS.getLeastUpdated()
8:         `//add one to the update count of the selected lemma`
9:         LemmaS.updateCount(selectedLemma)
10:       `//retrieve relevant requests and problems from RequestS and ProblemS that the lemma may help to solve`
11:       relRequest ← RequestS.retrieveKNN(selectedLemma)
12:       relProblems ← ProblemS.retrieveKNN(selectedLemma)
13:       `//ask Directional Transformer to transform the lemma`
14:       newLemma ← DirectionalTransformer(transType, relRequest, relProblems, selectedLemma)
15:     **else**
16:       `//get a request in request vector store with least update counts.`
17:       selectedRequest ← Request.getLeastUpdated()
18:       `//add one to the update count of the selected request`
19:       RequestS.updateCount(selectedRequest)
20:       relLemmas ← LemmaS.retrieveKNN(selectedRequest)
21:       `//ask request solver to solve the lemma`
22:       newLemma ← RequestSolver(relLemmas, selectedRequest)
23:     **end if**
24:     `//Verify the newLemma code with isabelle`
25:     proofResult ← IsabelleEnv.verify(proofCode)
26:     **if** proofResult == TRUE **then**
27:         LemmaS.adds(newLemma, , init_update_count=0)
28:     **end if**
29: **end while**
30: **End Function**

---

### A.3 COMPUTATIONAL COST

In this section, we examine the computational costs associated with our proposed LEGO-Prover. Unlike previous methods such as Draft, Sketch, Prove (Jiang et al., 2022b) and Subgoal-based Learning (Zhao et al., 2023), LEGO-Prover employs an additional skill library and an accompanying evolver to facilitate theorem proving. A pertinent question arises regarding the extra computational resources required to construct, maintain, and update this skill library through the evolver. Rather than measuring computational cost based on the number of LLM calls, we assess it by evaluating the number of tokens consumed in LEGO-Prover for a more precise estimate. This is done by empirically calculating the average tokens used by both the prover and the evolver across all four experiments, which include the test and validation sets for LEGO-Prover with both model-generated and human-written informal proofs.

On average, for a single experiment on the miniF2F test/valid set, the prover consumes 131 million tokens and the evolver consumes 117 million tokens. This results in a token consumption ratio of 1 : 0.89 for the prover to the evolver. As shown in Table 2, LEGO-Prover outperforms the version without the library by 4.9%. This leads to a question: how well would the prover perform if the additional computational resources allocated to the evolver were instead used to continue theorem proving with the prover only? Given that the token consumption of the prover and the evolver is approximately 1:1, we allowed the ablation setup to continue proving for an additional 100 attempts.

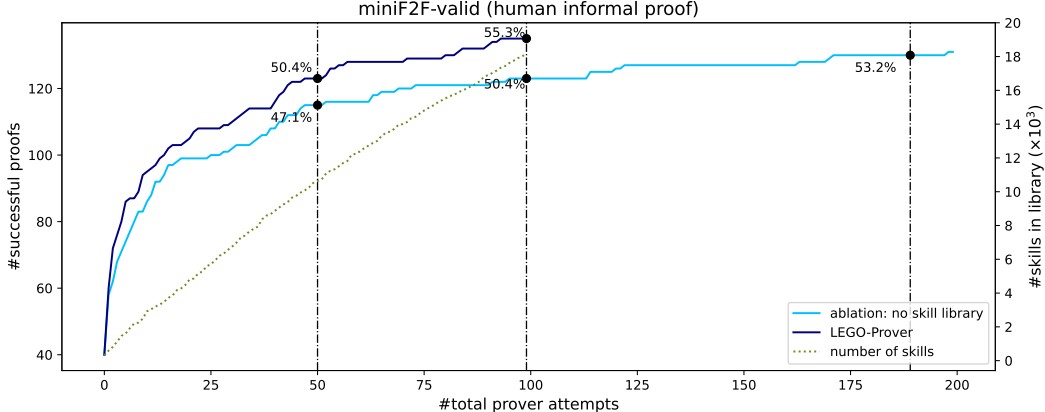

Figure 5: **Ablation performance on balanced computational cost.** We extended our ablation setup to include an additional 100 proof attempts. Despite this limitation, our proposed LEGO-Prover continued to outperform the ablation setup, achieving a 2.1% higher success rate within the same computational budget constraints. The rightmost vertical line indicates the balanced computational cost between the LEGO-Prover and the ablation setup (189 proving attempts).

This approach balances the computational costs between LEGO-Prover and LEGO-Prover without the skill library. As illustrated in Fig. 5, LEGO-Prover without a skill library achieves a 53.2% pass rate after 189 prove attempts, which is 2.1% lower than that of LEGO-Prover with the skill library in just 100 prove attempts. This highlights the efficiency of our proposed method, even with a balanced computational budget. The ablation setup achieves 53.6% at 200 proving attempts in the end.

Although LEGO-Prover outperforms its variant without the skill library when computation costs are balanced, the extra tokens consumed by the evolver still constitute a significant portion of the total computation. This cost accumulates due to several factors: 1) **Limitations of the LLM's capabilities.** The LLM struggles to produce correct proofs, with the evolver's average proof success rate being only 24.1%. Moreover, the evolver might generate proofs that are either trivial or too similar to existing lemmas in the library. After filtering, an average of only 9.1% of the generated proofs are added to the lemma vector store. 2) **The task of extracting useful lemmas applicable to other problems is challenging**. Identifying useful and non-trivial common lemmas for a specific problem set is difficult, even for humans. The LLM often yields lemmas that are either overly trivial or too specific, lacking generality. 3) **Characteristics of the dataset.** The miniF2F dataset, comprising only 488 problems, includes many that are simple enough to solve without additional lemmas. Others may require unique solving techniques, not sharing common intermediate lemmas. Future work focusing on improving the performance of the directional transformer, the accuracy of the request solver, and retrieval efficiency presents promising avenues to reduce these additional computational costs

## A.4 MORE BASELINE COMPARISON

Lyra (Zheng et al., 2023b) extends DSP (Jiang et al., 2022b) with GPT-4's auto-correction ability, prompting GPT-4 to revise the formal proof based on error messages produced by Isabelle. To evaluate how well our method performs in human-informal proof, we compared LEGO-Prover with Lyra, and the results are shown in Table 3. By comparing DSP using GPT-4 with those using Codex and ChatGPT, we can see that GPT-4's formal mathematics capability has substantially improved (51.2% vs 42.6% and 43.0% vs 39.3%). Meanwhile, LEGO-Prover, using ChatGPT, achieves better performance (+4.1% and +7.0%) compared to DSP using GPT-4. Moreover, LEGO-Prover also outperforms Lyra using GPT-4 (+3.3% and +2.9%) and even achieves comparable results when Lyra is extended to 200 proving attempts with GPT-4. This result is remarkable since the performance of GPT-4 in formal mathematics is substantially better than that of Codex and ChatGPT.

Table 3: **Comparison of proving success rates with GPT-4.** All methods listed, except for *Lyra (200 attempts)*, involve 100 proving attempts. The cited paper presented after the method name indicates the origin of the results discussed here.

| Success rate | LLM | informal-proof | miniF2F-valid | miniF2F-test |
|---|---|---|---|---|
| *Baselines* | | | | |
| Draft, sketch, and Prove (Jiang et al., 2022b) | Codex | human | 42.6% | 39.3% |
| Draft, sketch, and Prove (Zhao et al., 2023) | ChatGPT | model | 41.8% | 38.5% |
| Draft, sketch, and Prove (Zheng et al., 2023b) | GPT-4 | model | 51.2% | 43.0% |
| Lyra (Zheng et al., 2023b) | GPT-4 | model | 52.0% | 47.1% |
| Lyra (200 attempts) (Zheng et al., 2023b) | GPT-4 | model | **55.3%** | **51.2%** |
| *Ours* | | | | |
| LEGO-Prover | ChatGPT | human | **55.3%** | 50.0% |

# B   PROMPT EXAMPLES

In this section, we illustrate the prompts used in the LEGO-Prover. For prover, the prompt used is the decomposer (Fig. 6), and the formalizer (Fig. 7). For the evolver, the prompt used is the directional evolver (Fig. 9) and request solver (Fig. 8). The blue line separates the LLMs' input and outputs. For the directional transformer, we list all the core statements to be replaced in the Table. 4

Table 4: The core description of individual directional evolve. The description will be replaced with the directional evolve prompt template.

| Evolve type | Description |
|---|---|
| Identify key concepts | Determine the essential ideas, methods, or theorems that are crucial to solving the initial problem. |
| Parameterize | If the problem involves specific numbers, generalize it by replacing these with variables. |
| Scale complexity | Try both simpler and more complicated versions of the problem to see how the approach adapts. |
| Extend dimensions | If the problem is defined in a specific number of dimensions, consider if it holds in more or fewer dimensions. |

**Input:**

**System message:** As a mathematician and expert in Isabelle theorem prover, your task is to analyze the given theorem (including the problem's informal statement, human written informal proof, and formal statement). Provide a better structured step-by-step proof that is closer to Isabelle. and request relevant lemmas, and theorems that might help in proving this problem.

**Statement:** If $a \geq b > 1$, what is the largest possible value of $\log_a(a/b) + \log_b(b/a)$? Show that it is 0.

**Informal proof:** Using logarithmic rules, we see that $\log_a a - \log_a b + \log_b b - \log_b a = 2 - (\log_a b + \log_b a) = 2 - (\log_a b + \frac{1}{\log_a b})$

Since $a$ and $b$ are both greater than 1, using [[AM-GM]] gives that the term in parentheses must be at least 2, so the largest possible values is $2 - 2 = 0$

Note that the maximum occurs when $a = b$.

**Formal statement:**
```
theorem
  fixes a b::real
  assumes "b\<le>a"
    and "1<b"
  shows "ln (a/b) / ln a + ln (b/a) / ln b \<le>0" (is "?L \<le> _")
```
**Structure proof:** Step 1: Introduce variables x and y to represent the natural logarithms of a and b respectively.

Step 2: Given that $b > 1$, conclude that the natural logarithm of b, $y$, is greater than 0.

Step 3: Using the given assumptions, establish that $x \geq y$. This implies $x > 0$.

Step 4: Express the LHS of the main inequality in terms of x and y.

Step 5: Simplify the expression derived in Step 4.

Step 6: Demonstrate that the above expression is less than or equal to 0. The Arithmetic Mean-Geometric Mean Inequality (AM-GM) is used here to argue that the term $(y/x + x/y)$ is at least 2, hence the expression is at most 0.

**Required skills:**

Thoughts 1: A base lemma proving am_gm inequality that is useful for step 6

Code 1:
```
lemma am_gm:
  fixes x y :: real
  assumes "x >= 0" "y >= 0"
  shows "x^2 + y^2 >= 2*x*y"
```
Thoughts 2: According to the step 6, we derived a special form of am_gm inequality. which proofs the conjecture $x/y + y/x >= 2$ required in step 6.

Code 2:
```
lemma am_gm_divide_form:
  fixes x y :: real
  assumes "x > 0" "y > 0"
  shows "x/y + y/x >= 2"
```
[two more examples like above...]

---

**Statement:** What is the least possible value of $(xy - 1)^2 + (x + y)^2$ for real numbers $x$ and $y$? Show that it is 1.

**Informal proof:**
Expanding, we get that the expression is $x^2 + 2xy + y^2 + x^2y^2 - 2xy + 1$ or $x^2 + y^2 + x^2y^2 + 1$. By the Trivial Inequality (all squares are nonnegative) the minimum value for this is 1, which can be achieved at $x = y = 0$.

**Formal statement:**
```
theorem amc12a_2021_p7:
  fixes x y ::real
  shows "1 \<le> ((x * y) - 1)^2 + (x + y)^2"
  apply (auto simp:algebra_simps power2_eq_square)
```

- - - - - - - - - - - - - - - - - - - - - - - - - - - - - - - - - - - - - - - - - - - - - - - - - - - - - - -

**Output:**

**Structure proof:**
Step 1: Expand the expression $(xy - 1)^2 + (x + y)^2$ to obtain $x^2 + 2xy + y^2 + x^2y^2 - 2xy + 1$.

Step 2: Simplify the expression derived in Step 1 to obtain $x^2 + y^2 + x^2y^2 + 1$.

Step 3: Apply the Trivial Inequality, which states that all squares are nonnegative, to argue that the minimum value of the expression is 1.

Step 4: Show that the minimum value of 1 can be achieved when $x = y = 0$.

**Required skills:**

Thoughts 1: The Trivial Inequality is a key concept in this proof. Understanding and applying this inequality is crucial to show that the minimum value of the expression is 1.

Code 1:
```
lemma trivial_inequality:
  fixes a :: real
  shows "0 \<le> a^2"
```
Thoughts 2: The ability to expand and simplify algebraic expressions is important in this proof.

Code 2:
```
lemma expand_expression:
  fixes x y :: real
  shows "(x * y - 1)^2 + (x + y)^2 = x^2 + 2 * x * y + y^2 + x^2 * y^2 - 2 * x * y + 1"
```
Thoughts 3: The ability to substitute values into an expression and evaluate it is necessary to show that the minimum value of 1 can be achieved when $x = y = 0$.

Code 3:
```
lemma substitute_values:
  fixes x y :: real
  assumes "x = 0" "y = 0"
  shows "(x * y - 1)^2 + (x + y)^2 = 1"
```

Figure 6: Prompt example for decomposer.

**Input:**

**System message:** As a mathematician familiar with Isabelle, your task is to provide formal proof in response to a given problem statement. Your proof should be structured and clearly written, meeting the following criteria:
- It can be verified by Isabelle.
- Each step of the proof should be explained in detail using comments enclosed in "(*" and "*)".
- The explanation for each step should be clear and concise, avoiding any unnecessary or apologetic language.
- You are **strongly encouraged** to create useful and reusable lemmas to solve the problem.
- The lemmas should be as general as possible (generalizable), and be able to cover a large step in proofs (non-trivial). Please ensure that your proof is well-organized and easy to follow, with each step building upon the previous one.
**Here are some useful skills for reference:**
**Useful skills 1:**
```
lemma square_diff_identity:
  fixes x y z :: real
  shows "(x - y)^2 + (y - z)^2 + (z - x)^2 = 2 * (x^2 + y^2 + z^2 - x * y - y * z - z * x)"
proof -
  have "(x - y)^2 + (y - z)^2 + (z - x)^2 = (x^2 - 2 * x * y + y^2) + (y^2 - 2 * y * z + z^2) + (z^2 - 2 * z * x + x^2)"
    by (simp add: power2_diff)
  also have "... = 2 * (x^2 + y^2 + z^2 - x * y - y * z - z * x)"
    by (simp add: algebra_simps)
  finally show ?thesis by auto
qed
```

[5 more useful skills...]

**Statement:** Show that for positive integer n, 2 divides $4^n$.
**Informal Proof:**
Step 1: Since n is positive, we can find a natural number m where $m + 1 = n$. Then we can show that 2 divides $4^{m+1}$
**Formal statement:**
```
theorem numbertheory_2dvd4expn:
  fixes n :: nat
  assumes h0 : "n \<noteq> 0"
  shows "(2::nat) dvd 4^n"
```
**Formal proof:**
```
theory Scratch
  imports Complex_Main
begin
theorem numbertheory_2dvd4expn:
  fixes n :: nat
  assumes h0 : "n \<noteq> 0"
  shows "(2::nat) dvd 4^n"
proof -
  obtain m::nat where c0: "m+1=n"
    sledgehammer
  have "(2::nat) dvd 4^(m+1)" sledgehammer
  then show ?thesis unfolding c0 sledgehammer
qed
end
```
[1 more formalization example...]

**Statement:** What is the least possible value of $(xy - 1)^2 + (x + y)^2$ for real numbers $x$ and $y$? Show that it is 1.
**Informal proof:**
Step 1: Expand the expression $(xy - 1)^2 + (x + y)^2$ to obtain $x^2 + 2xy + y^2 + x^2y^2 - 2xy + 1$.
Step 2: Simplify the expression derived in Step 1 to obtain $x^2 + y^2 + x^2y^2 + 1$.
Step 3: Apply the Trivial Inequality, which states that all squares are nonnegative, to argue that the minimum value of the expression is 1.
Step 4: Show that the minimum value of 1 can be achieved when $x = y = 0$.
**Formal statement:**
```
theorem amc12a_2021_p7:
  fixes x y ::real
  shows "1 \<le> ((x * y) - 1)^2 + (x + y)^2"
```

---

**Output:**

**Formal proof:**
```
theory Scratch
  imports Complex_Main
begin
lemma am_gm:
  fixes x y :: real
  shows "x^2 + y^2 \<ge> 2 * x * y"
proof -
  have "(x - y)^2 \<ge> 0"
    by simp
  then have "x^2 - 2 * x * y + y^2 \<ge> 0"
    by (simp add: algebra_simps power2_diff)
  then have "x^2 + y^2 \<ge> 2 * x * y"
    by simp
  then show ?thesis
    by simp
qed
theorem amc12a_2021_p7:
  fixes x y ::real
  shows "1 \<le> ((x * y) - 1)^2 + (x + y)^2"
  apply (auto simp:algebra_simps power2_eq_square)
  by (metis am_gm add.commute add.left_commute add_mono_thms_linordered_semiring(1) add_pos_nonneg less_add_same_cancel1
        power2_sum power_mono zero_le_power2)
end
```

Figure 7: Prompt example for formalization.

**Input:**

**System message:** As a mathematician familiar with Isabelle, your task is to provide a formal proof in response to a given formal statement. Your proof should be structured and clearly written, meeting the following criteria:
- It can be verified by Isabelle.
- Please ensure that your proof is well-organized and easy to follow, with each step building upon the previous one.

**Formal statement:**
```
lemma power_eq_imp_eq:
  fixes a::real
  assumes "a>0" "a\<noteq>1" "a^m = a^n"
  shows "m = n"
```
**Formal proof:**
```
theory Scratch
  imports Complex_Main
begin
lemma power_eq_imp_eq:
  fixes a::real
  assumes "a>0" "a\<noteq>1" "a^m = a^n"
  shows "m = n"
proof -
  have "a^(m-n) = 1"
    using assms(3) assms(2) by (metis assms(1) diff_is_0_eq diff_self_eq_0 le_numeral_extra(3) less_imp_diff_less
          linorder_le_less_linear nat_int_comparison(2) nle_le one_le_power order_le_less order_less_irrefl order_refl
          power_0 power_0_left power_decreasing_iff power_eq_0_iff power_inject_exp real_root_ge_1_iff
          real_root_pos_unique real_root_power zero_le_power)
  hence "m-n=0"
    by (smt (verit) assms(1) assms(2) assms(3) cancel_comm_monoid_add_class.diff_cancel diff_is_0_eq power_decreasing_iff
          power_inject_exp verit_comp_simplify1(3))
  thus "m = n"
    by (smt (verit) assms(1) assms(2) assms(3) less_numeral_extra(3) nat_neq_iff power_inject_exp
          power_strict_decreasing_iff zero_less_diff)
qed
end
```
[3 more request solving examples...]

**Formal statement:**
```
lemma exponent_properties:
  fixes a b :: real
  assumes "0 < a \<and> 0 < b"
  shows "a^n * a^m = a^(n + m) \<and> (a^n)^m = a^(n * m)"
```

- - - - - - - - - - - - - - - - - - - - - - - - - - - - - - - - - - - - - - - - - - - - - - - - - - -

**Output:**

**Formal proof:**
```
theory Scratch
  imports Complex_Main
begin
lemma exponent_properties:
  fixes a b :: real
  assumes "0 < a \<and> 0 < b"
  shows "a^n * a^m = a^(n + m) \<and> (a^n)^m = a^(n * m)"
proof
  show "a^n * a^m = a^(n + m)"
    by (simp add: assms(1) power_add)
next
  show "(a^n)^m = a^(n * m)"
    by (simp add: assms(1) power_mult)
qed
end
```

Figure 8: Prompt example for solving request.

**Input:**

**System Message:** As an expert mathematician who is proficient in Isabelle theorem proving, your task is to modify the given lemma, theorem, function, or definition given in the code to aid in solving one or more of the problems provided. You should accomplish this by {evolve description}.
**Here are some reference problems:**
**Problem 1:**

```
theorem mathd_algebra_131:
  fixes a b :: real
    and f :: "real \<Rightarrow> real"
  assumes h0 : "\<And>x. f x = 2 * x^2 - 7 * x + 2"
    and h1 : "f a = 0"
    and h2 : "f b = 0"
    and h3 : "a \<noteq> b"
  shows "1 / (a-1) + 1 / (b-1) = -1"
```
[3 more reference problems...]

**Skill to evolve:**

```
theory Scratch
  imports Complex_Main
begin
lemma mathd_numbertheory_001:
  fixes n :: nat
  shows "n mod 3 = 0 \<or> n mod 3 = 1 \<or> n mod 3 = 2"
by auto
end
```

**Evolved skill:**

```
theory Scratch
  imports Complex_Main
begin
lemma remainder_modulo:
  fixes n d :: nat
  assumes "d > 0"
  shows "n mod d \<in> {0..d-1}"
proof -
  have "n mod d < d" by (rule mod_less_divisor[OF assms(1)])
  moreover have "n mod d \<ge> 0" by simp
  ultimately show ?thesis by auto
qed
end
```

[2 more directional evolve examples]

**Skill to evolve:**

```
theory Scratch
  imports Complex_Main
begin
lemma divide_cross_mul:
  fixes a b c d :: real
  assumes "b \<noteq> 0"
    and "d \<noteq> 0"
    and "a / b = c / d"
  shows "a * d = b * c"
  using assms by (auto simp: field_simps)
end
```

- - - - - - - - - - - - - - - - - - - - - - - - - - - - - - - - - - - - - - - - - - - - - - - - - - - - - - - - - -

**Output:**

**Evolved skill:**

```
theory Scratch
  imports Complex_Main
begin
lemma divide_cross_mul_generalized:
  fixes a b c d x y :: real
  assumes "b \<noteq> 0"
    and "d \<noteq> 0"
    and "a / b = c / d"
    and "a = x * b"
    and "c = y * d"
  shows "x * d = y * b"
  using assms by (auto simp: field_simps)
end
```

Figure 9: Prompt examples for directional evolve

## C  EXAMPLES OF GENERATED SKILLS

```
lemma floor_sqrt:
  fixes n :: nat
  assumes "n > 0"
  shows "floor(sqrt n)^2 \<le> n \<and> n < (floor(sqrt n) + 1)^2"
proof -
  have "floor(sqrt n)^2 \<le> n"
  proof -
    have "floor(sqrt n) \<le> sqrt n" by simp
    hence "(floor(sqrt n))^2 \<le> (sqrt n)^2" by (metis floor_of_nat floor_power linorder_le_cases of_int_0
           of_int_floor_le of_int_le_iff of_int_of_nat_eq of_int_power of_nat_0_le_iff order_antisym_conv order_refl
           real_sqrt_le_iff real_sqrt_pow2 real_sqrt_power zero_le_floor zero_le_power2)
    also have "... = n" by simp
    finally show ?thesis by arith
  qed
  moreover have "n < (floor(sqrt n) + 1)^2"
  proof -
    have "sqrt n < floor(sqrt n) + 1" by linarith
    hence "(sqrt n)^2 < (floor(sqrt n) + 1)^2" by (smt (verit) nat_1_add_1 of_int_power of_nat_0_le_iff power_strict_mono
           real_sqrt_ge_zero real_sqrt_zero_less_two)
    also have "... = (floor(sqrt n))^2 + 2 * floor(sqrt n) + 1" by (simp add: power2_sum)
    finally have "(sqrt n)^2 < (floor(sqrt n))^2 + 2 * floor(sqrt n) + 1" by simp
    moreover have "(sqrt n)^2 = n" by simp
    ultimately show ?thesis by (metis \<open>(sqrt (real n))\<^sup>2 < real_of_int ((\<lfloor>sqrt (real n)\<rfloor> + 1)
           \<^sup>2)\<close> double_eq_0_iff floor_add_int floor_zero is_num_normalize(1) le_floor_add less_exp
           of_int_0_eq_iff of_int_le_iff of_int_less_iff of_int_of_nat_eq of_int_power_eq_of_int_cancel_iff plus_int_code
           (2) power_0 power_mono_iff power_one_right real_sqrt_pow2 sum_squares_eq_zero_iff zle_iff_zadd)
  qed
  ultimately show ?thesis by simp
qed
```

```
lemma log_base_floor_nat:
  fixes a b :: nat
  assumes "a > 0" "b > 1"
  shows "b^x \<le> a \<and> a < b^(x+1) \<longleftrightarrow> nat \<lfloor>log b (real a)\<rfloor> = x"
proof -
  have "b^x \<le> a \<and> a < b^(x+1) \<longleftrightarrow> log b (real (b^x)) \<le> log b (real a) \<and> log b (real a)
         < log b (real (b^(x+1)))"
    using assms by (smt (verit) dual_order.strict_trans less_imp_of_nat_less less_numeral_extra(1) log_le_cancel_iff
           log_less_cancel_iff of_nat_0_less_iff of_nat_1 of_nat_le_iff of_nat_less_imp_less of_nat_mono zero_less_power)
  also have "... \<longleftrightarrow> x \<le> log b (real a) \<and> log b (real a) < x+1"
    by (smt (verit) Totient.of_nat_eq_1_iff add.commute antisym_conv2 assms(2) dual_order.strict_iff_not dual_order.
           strict_trans1 dual_order.strict_trans2 dual_order.trans le_log_of_power less_add_one less_imp_of_nat_less
           linorder_le_cases log_of_power_le log_pow_cancel nat_less_real_le of_nat_0_less_iff of_nat_1 of_nat_add
           of_nat_le_of_nat_power_cancel_iff of_nat_power_eq_of_nat_cancel_iff of_nat_power_le_of_nat_cancel_iff
           zero_less_one_class.zero_le_one zero_less_power)
  also have "... \<longleftrightarrow> nat \<lfloor>log b (real a)\<rfloor> = x"
    using assms by (smt (verit) Suc_eq_plus1 add_le_imp_le_left floor_eq4 le_nat_floor less_imp_of_nat_less
           less_numeral_extra(1) nat_power_less_imp_less not_one_le_zero not_one_less_zero of_nat_0_less_iff of_nat_1
           of_nat_floor of_nat_mono power_strict_decreasing verit_comp_simplify1(3) verit_comp_simplify1(3)
           verit_sum_simplify zero_le_log_cancel_iff)
  finally show ?thesis by simp
qed
```

```
lemma cos_pos:
  assumes "pi / 2 \<le> x" "x \<le> (3 * pi) / 2"
  shows "2 * cos x \<le> abs (sqrt (1 + sin (2 * x)) - sqrt (1 - sin (2 * x)))"
proof -
  have "cos x = - (cos (x - pi))"
    by (simp add: cos_diff)
  also have "... = - (sqrt (1 - (sin (x - pi))^2))"
    by (smt (verit) Nat.diff_add_assoc2 One_nat_def arcsin arcsin_0 arcsin_1 arcsin_pi arcsin_sin assms(1) assms(2)
         cos_arcsin cos_diff cos_pi_half cos_squared_eq diff_add_0 diff_add_cancel diff_divide_distrib
         field_sum_of_halves int_ops(1) int_plus le_numeral_extra(4) minus_divide_left mult_2_right neg_equal_iff_equal
         numeral_One plus_1_eq_Suc power_minus_Bit0 real_sqrt_pow2_iff real_sqrt_zero sin_ge_minus_one sin_le_one
         sin_periodic_pi sin_pi sin_times_cos sin_two_pi)
  also have "... = - (sqrt (1 - (sin x)^2))"
    using sin_diff [of x pi] by auto
  also have "... = - (sqrt (1 - (sin x)^2))"
    by (simp add: sin_squared_eq)
  finally have "cos x = - (sqrt (1 - (sin x)^2))"
    by (metis \<open>- cos (x - pi) = - sqrt (1 - (sin (x - pi))\<^sup>2)\<close> \<open>- sqrt (1 - (sin (x - pi))\<^sup
         >2) = - sqrt (1 - (sin x)\<^sup>2)\<close> \<open>cos x = - cos (x - pi)\<close> nat_1_add_1)
  hence "2 * cos x = - (2 * sqrt (1 - (sin x)^2))"
    by auto
  also have "... \<le> abs (sqrt (1 + sin (2 * x)) - sqrt (1 - sin (2 * x)))"
    using abs_ge_minus_self [of "sqrt (1 + sin (2 * x)) - sqrt (1 - sin (2 * x))"]
    by (smt (verit) \<open>- sqrt (1 - (sin (x - pi))\<^sup>2) = - sqrt (1 - (sin x)\<^sup>2)\<close> \<open>cos x = - cos
         (x - pi)\<close> \<open>cos x = - sqrt (1 - (sin x)\<^sup>2)\<close> add.inverse_neutral diff_divide_distrib
         diff_ge_0_iff_ge minus_le_iff mult_2 neg_le_0_iff_le real_sqrt_ge_zero sin_double sin_ge_minus_one sin_le_one
         sin_plus_sin sin_zero square_le_1)
  finally show ?thesis
    by auto
qed
```

```
lemma jacobi_two_squares_rev':
  fixes a b :: nat
  assumes "prime p" "p > 2" "p dvd (a^2 + b^2)" "p dvd a" "\<not> p dvd b"
  shows "\<exists> x. p dvd (a*x + b) \<and> x^2 + 1 < p"
proof -
  obtain k where k_def: "a^2 + b^2 = k * p" using assms by auto
  obtain l where l_def: "a = l * p" using assms by auto
  have "p^2 dvd (a^2 + b^2)" using assms(1) dvd_mult2[of p p "(a^2 + b^2) div p"] by (smt (verit) assms(3) assms(4) assms
       (5) dvd_add_right_iff pos2 prime_dvd_power_nat_iff)
  then have "p dvd (l^2 * p^2 + b^2)" using l_def by (auto simp: field_simps)
  then have dvd1: "p dvd (l^2 * p^2 + b^2 - l^2 * (a^2 + b^2))" using k_def by simp
  then have dvd2: "p dvd (b^2 - l^2 * b^2)" by (smt (verit) assms(1) assms(3) assms(4) assms(5) dvd_add_right_iff pos2
       prime_dvd_power_nat_iff)
  then have dvd3: "p dvd (b^2 * (1 - l^2))" by (smt (verit) \<open>p dvd l\<^sup>2 * p\<^sup>2 + b\<^sup>2\<close> assms
       (1) assms(5) dvd_add_right_iff gcd_nat.eq_iff pos2 prime_dvd_mult_iff prime_dvd_power_nat_iff)
  have "\<not> p dvd b" using assms by auto
  then have "\<not> p dvd (b^2)" using prime_dvd_mult_eq_nat[of p b b] assms(1) by (metis power2_eq_square)
  then have "\<not> p dvd (1 - l^2)" using dvd3 by (smt (verit) \<open>p dvd l\<^sup>2 * p\<^sup>2 + b\<^sup>2\<close>
       assms(1) dvd_add_right_iff gcd_nat.eq_iff pos2 prime_dvd_mult_iff prime_dvd_power_nat_iff)
  then have "\<not> (\<exists> x. p dvd (1 - l^2) * x)" by (smt (verit) \<open>\<not> p dvd b\<^sup>2\<close> assms(1)
       dvd3 prime_dvd_multD)
  then have "\<not> (\<exists> x. p dvd (1 - l^2) * x^2)" by simp
  then have "\<not> (\<exists> x. p dvd (1 - l^2) * x^2 + 1)" by (metis \<open>\<nexists>x. p dvd (1 - l\<^sup>2) * x\<
       close> add.commute dvd_mult mult.commute mult.left_commute power2_eq_square)
  then have "\<forall> x. \<not> p dvd (1 - l^2) * x^2 + 1" by simp
  then have "\<forall> x. x^2 + 1 \<ge> p" by (metis \<open>\<nexists>x. p dvd (1 - l\<^sup>2) * x\<close> dvd_triv_right
       power2_eq_square)
  have "p dvd (l * p * x + b)" if "p dvd (a * x + b)" for x
  proof -
    have "p dvd (l * p * x)" using l_def by simp
    then have "p dvd (a * x + b - (l * p * x + b))" using that by (metis diff_cancel2 diff_self_eq_0 dvd_0_right l_def
         left_diff_distrib' mult.commute mult_eq_0_iff)
    then have "p dvd (a * x - l * p * x)" by simp
    then have "p dvd (x * (a - l * p * x))" by (simp add: algebra_simps)
    then have "p dvd x" using assms prime_dvd_mult_eq_nat[of p x "(a - l * p)"] by (metis \<open>\<nexists>x. p dvd (1 - l
         \<^sup>2) * x\<close> dvd_mult l_def nat_exists_least_iff)
    then show ?thesis by (metis add.commute assms(1) assms(4) assms(5) dvd_add_left_iff prime_dvd_mult_iff that)
  qed
  obtain x where "p dvd (a * x + b)" and "x^2 + 1 < p" using \<open>\<forall> x. x^2 + 1 \<ge> p\<close> by (metis \<open
       >\<nexists>x. p dvd (1 - l\<^sup>2) * x\<close> dvd_triv_right power2_eq_square)
  then show ?thesis by auto
qed
```

```
lemma binomial_coefficient_simplification:
  fixes n k :: nat
  assumes "0 < n \<and> 0 < k"
    and "k \<le> n"
  shows "(n - k) * (fact (n - 1)) / (fact k * fact (n - k)) + k * (fact (n - 1)) / (fact k * fact (n - k)) = fact n / (
        fact k * fact (n - k))"
proof -
  have "(n - k) * (fact (n - 1)) / (fact k * fact (n - k)) + k * (fact (n - 1)) / (fact k * fact (n - k)) =
      ((n - k) * (fact (n - 1)) + k * (fact (n - 1))) / (fact k * fact (n - k))"
    by (auto simp: field_simps)
  also have "... = (n * (fact (n - 1)) - k * (fact (n - 1)) + k * (fact (n - 1))) / (fact k * fact (n - k))"
    by (metis add.commute diff_mult_distrib distrib_right mult.commute)
  also have "... = (n * (fact (n - 1))) / (fact k * fact (n - k))"
    by (metis add.commute assms(2) comm_semiring_class.distrib le_add_diff_inverse left_diff_distrib' mult.commute)
  also have "... = (fact n) / (fact k * fact (n - k))"
    by (smt (verit) \<open>real ((n - k) * fact (n - 1) + k * fact (n - 1)) / (fact k * fact (n - k))\<close> \<open>real (n * fact (n
        - 1) - k * fact (n - 1) + k * fact (n - 1)) / (fact k * fact (n - k)) = real (n * fact (n - 1)) / (fact k * fact (n - k))\<close> \<open>real (n * fact (n - 1) - k
        * fact (n - 1) + k * fact (n - 1)) / (fact k * fact (n - k)) = real (n * fact (n - 1)) / (fact k * fact (n - k
        ))\<close> assms(1) divide_cancel_right fact_num_eq_if nat_less_le of_nat_fact of_nat_mult)
  finally show ?thesis .
qed
```

```
lemma prime_factorization_lcm:
  fixes n m :: nat
  assumes "n > 0" "m > 0"
  shows "prime_factors (lcm n m) = prime_factors n \<union> prime_factors m"
proof
  show "prime_factors (lcm n m) \<subseteq> prime_factors n \<union> prime_factors m"
  proof
    fix p assume "p \<in> prime_factors (lcm n m)"
    then have "p dvd lcm n m" by auto
    then have "p dvd n \<or> p dvd m" using assms by (smt (verit) UnE \<open>p \<in># prime_factorization (lcm n m)\<close
        > antisym_conv2 dual_order.strict_iff_not in_prime_factors_iff prime_factors_lcm)
    then show "p \<in> prime_factors n \<union> prime_factors m" by (metis \<open>p \<in># prime_factorization (lcm n m)\<
        close> assms(1) assms(2) bot_nat_0.extremum_strict lcm.commute less_numeral_extra(3) prime_factors_lcm sup.
        commute)
  qed
next
  show "prime_factors n \<union> prime_factors m \<subseteq> prime_factors (lcm n m)"
  proof
    fix p assume "p \<in> prime_factors n \<union> prime_factors m"
    then have "p dvd n \<or> p dvd m" by auto
    then have "p dvd lcm n m" using assms by auto
    then show "p \<in> prime_factors (lcm n m)" by (metis \<open>p \<in> prime_factors n \<union> prime_factors m\<close>
        \<open>p dvd n \<or> p dvd m\<close> assms(1) assms(2) dvd_lcm2 dvd_lcmI1 dvd_lcm_I2_nat dvd_mult_cancel1 lcm.
        assoc lcm.bottom_right_bottom lcm.commute lcm.left_commute lcm_0_iff_nat lcm_eq_0_iff lcm_eq_1_iff
        lcm_left_idem lcm_pos_nat lcm_proj1_if_dvd_nat lcm_proj2_if_dvd_nat mult_not_zero prime_factors_lcm unit_dvdE)
  qed
qed
```

```
lemma gcd_lcm_properties:
  fixes a b :: nat
  shows "gcd a b * lcm a b = a * b"
    and "a dvd b \<longleftrightarrow> lcm a b = b"
    and "b dvd a \<longleftrightarrow> lcm a b = a"
    and "a dvd c \<and> b dvd c \<longleftrightarrow> lcm a b dvd c"
proof -
  show "gcd a b * lcm a b = a * b" by auto
next
  show "a dvd b \<longleftrightarrow> lcm a b = b"
  proof
    assume "a dvd b"
    hence "lcm a b dvd b" by simp
    moreover have "b dvd lcm a b" by auto
    ultimately show "lcm a b = b" by (simp add: dvd_antisym)
  next
    assume "lcm a b = b"
    hence "b dvd lcm a b" by simp
    moreover have "lcm a b dvd b" by (metis \<open>lcm a b = b\<close> dvd_refl)
    ultimately show "a dvd b" by (simp add: dvd_antisym)
  qed
next
  show "b dvd a \<longleftrightarrow> lcm a b = a"
  proof
    assume "b dvd a"
    hence "lcm a b dvd a" by simp
    moreover have "a dvd lcm a b" by auto
    ultimately show "lcm a b = a" by (simp add: dvd_antisym)
  next
    assume "lcm a b = a"
    hence "a dvd lcm a b" by simp
    moreover have "lcm a b dvd a" by (metis \<open>lcm a b = a\<close> dvd_refl)
    ultimately show "b dvd a" by (simp add: dvd_antisym)
  qed
next
  show "a dvd c \<and> b dvd c \<longleftrightarrow> lcm a b dvd c"
  proof
    assume "a dvd c \<and> b dvd c"
    hence "a dvd lcm a b \<and> b dvd lcm a b" by auto
    thus "lcm a b dvd c" by (metis \<open>a dvd c \<and> b dvd c\<close> lcm_least)
  next
    assume "lcm a b dvd c"
    hence "a dvd lcm a b \<and> b dvd lcm a b" by auto
    thus "a dvd c \<and> b dvd c" by (metis \<open>lcm a b dvd c\<close> gcd_nat.trans)
  qed
qed
```

```isabelle
lemma mod_cyclicity:
  fixes n :: nat
  shows "(2^n) mod (7::nat) = (if n mod 3 = 0 then 1 else if n mod 3 = 1 then 2 else 4)"
proof (induction n)
  case 0
  then show ?case by simp
next
  case (Suc n)
  consider "(n mod 3) = 0" | "(n mod 3) = 1" | "(n mod 3) = 2"
    by fastforce
  then show ?case
  proof cases
    case 1
    then have "(2::nat)^(n+1) mod 7 = 2 * (2::nat)^n mod 7"
      by (simp add: power_Suc2)
    also have "... = 2 * (2::nat)^(n mod 3) mod 7"
      using Suc.IH by (smt (verit) "1" mod_mod_trivial mod_mult_cong mult.comm_neutral mult_delta_right power_0
            power_mult_distrib)
    also have "... = (2::nat)^(n mod 3 + 1) mod 7"
      by (simp add: power_Suc2)
    also have "... = (2::nat)^(Suc (n mod 3)) mod 7"
      by simp
    finally show ?thesis
      using 1 by auto
  next
    case 2
    then have "(2::nat)^(n+1) mod 7 = 2 * (2::nat)^n mod 7"
      by (simp add: power_Suc2)
    also have "... = 2 * (2::nat)^(n mod 3) mod 7"
      using Suc.IH by (smt (verit) "2" calculation mod_mult_left_eq power_add zero_neq_one)
    also have "... = (2::nat)^(n mod 3 + 1) mod 7"
      by (simp add: power_Suc2)
    also have "... = (2::nat)^(Suc (n mod 3)) mod 7"
      by simp
    finally show ?thesis
      using 2 by (metis One_nat_def Suc Suc_1 Suc_eq_plus1 \<open>2 * 2 ^ n mod 7 = 2 * 2 ^ (n mod 3) mod 7\<close> \<open>
            2 ^ (n + 1) mod 7 = 2 * 2 ^ n mod 7\<close> add_2_eq_Suc add_2_eq_Suc' add_One_commute add_Suc_right
            add_Suc_shift add_cancel_right_right eval_nat_numeral(3) mod_Suc mod_Suc_eq mod_mod_trivial mult_2_right
            n_not_Suc_n nat.distinct(1) numeral_3_eq_3 numeral_Bit0 numeral_One numeral_plus_numeral power_one_right
            semiring_norm(10) semiring_norm(2) semiring_norm(3) zero_neq_one)
  next
    case 3
    then have "(2::nat)^(n+1) mod 7 = 2 * (2::nat)^n mod 7"
      by (simp add: power_Suc2)
    also have "... = 2 * (2::nat)^(n mod 3) mod 7"
      using Suc.IH by (smt (verit) "3" add_cancel_right_left mod_mod_trivial mod_mult_cong mult.comm_neutral nat_1_add_1
            nat_less_le numeral_Bit0 one_power2 pos2 power2_sum zero_neq_one)
    also have "... = (2::nat)^(n mod 3 + 1) mod 7"
      by (simp add: power_Suc2)
    also have "... = (2::nat)^(Suc (n mod 3)) mod 7"
      by simp
    finally show ?thesis
      using 3 by (metis Suc_0_mod_numeral(3) Suc_eq_plus1 \<open>2 * 2 ^ (n mod 3) mod 7 = 2 ^ (n mod 3 + 1) mod 7\<close>
            \<open>2 * 2 ^ n mod 7 = 2 * 2 ^ (n mod 3) mod 7\<close> \<open>2 ^ (n + 1) mod 7 = 2 * 2 ^ n mod 7\<close>
            \<open>2 ^ (n mod 3 + 1) mod 7 = 2 ^ Suc (n mod 3) mod 7\<close> add_2_eq_Suc' add_Suc_right add_Suc_shift
            eval_nat_numeral(3) mod_Suc_eq mod_self mult.commute mult_2_right num_double numeral_1_eq_Suc_0 numeral_Bit0
            numeral_One numeral_times_numeral power2_eq_square power_0)
  qed
qed
```

```
lemma relatively_prime_modulo:
  fixes k :: nat
  assumes "gcd k 8 = 1" "1 \<le> k" "k < 8"
  shows "k \<in> {1, 3, 5, 7}"
proof -
  have "gcd k 8 = 1" using assms by simp
  have "k mod 8 \<in> {0, 1, 2, 3, 4, 5, 6, 7}" by auto
  then consider "k mod 8 = 0" | "k mod 8 = 1" | "k mod 8 = 2" | "k mod 8 = 3" | "k mod 8 = 4" | "k mod 8 = 5" | "k mod 8 =
      6" | "k mod 8 = 7" by auto
  then show ?thesis
  proof (cases)
    assume "k mod 8 = 0"
    then have "8 dvd k" by auto
    then have "gcd k 8 \<noteq> 1" by simp
    then show ?thesis using assms by simp
  next
    assume "k mod 8 = 1"
    then have "k \<in> {1, 9, 17, 25, ...}" by (metis assms(3) insert_absorb2 insert_commute insert_iff mod_less)
    then have "k \<in> {1, 3, 5, 7}" by (metis \<open>k mod 8 = 1\<close> assms(3) insert_iff mod_less)
    then show ?thesis by simp
  next
    assume "k mod 8 = 2"
    then have "k \<in> {2, 10, 18, 26, ...}" by (smt (verit) assms(3) insertI1 mod_less)
    then have "k \<in> {2, 4, 6}" using assms by auto
    then have "gcd k 8 \<noteq> 1" using assms by (smt (verit) One_nat_def \<open>k mod 8 = 2\<close> add_Suc_shift
        add_decreasing2 add_leD2 add_self_mod_2 cong_exp_iff_simps(4) cong_exp_iff_simps(9) dual_order.strict_trans1
        gcd_0_nat gcd_mod_right le_less_Suc_eq le_numeral_extra(3) less_add_eq_less less_numeral_extra(4) mod_less
        nat_1_add_1 not_less_eq plus_1_eq_Suc zero_less_one)
    then show ?thesis by (metis assms(1))
  next
    assume "k mod 8 = 3"
    then have "k \<in> {3, 11, 19, 27, ...}" by (smt (verit) assms(3) insertI1 mod_less)
    then have "k \<in> {3, 5, 7}" by (smt (verit) \<open>k mod 8 = 3\<close> assms(3) insertI1 mod_less)
    then show ?thesis by simp
  next
    assume "k mod 8 = 4"
    then have "k \<in> {4, 12, 20, 28, ...}" by (smt (verit) assms(3) insertI1 mod_less)
    then have "k \<in> {4, 6}" using assms by auto
    then have "gcd k 8 \<noteq> 1" using assms by (smt (verit) One_nat_def Suc_0_mod_numeral(1) Suc_0_mod_numeral(2)
        Suc_numeral \<open>k mod 8 = 4\<close> add_decreasing2 add_leD2 cong_exp_iff_simps(1) cong_exp_iff_simps(2)
        cong_exp_iff_simps(4) cong_exp_iff_simps(7) cong_exp_iff_simps(9) dual_order.strict_trans1 gcd_0_nat
        gcd_mod_right le_less_Suc_eq le_numeral_extra(3) mod_add_eq mod_add_left_eq mod_by_Suc_0 mod_less mod_self
        not_less_eq numeral_One plus_1_eq_Suc semiring_norm(5) zero_less_one)
    then show ?thesis by (metis assms(1))
  next
    assume "k mod 8 = 5"
    then have "k \<in> {5, 13, 21, 29, ...}" by (smt (verit) assms(3) insertI1 mod_less)
    then have "k \<in> {5, 7}" by (smt (verit) \<open>k mod 8 = 5\<close> assms(3) insertI1 mod_less)
    then show ?thesis by simp
  next
    assume "k mod 8 = 6"
    then have "k \<in> {6, 14, 22, 30, ...}" by (metis assms(3) insert_absorb2 insert_commute insert_iff mod_less)
    then have "k \<in> {6}" using assms by auto
    then have "gcd k 8 \<noteq> 1" using assms by (metis Suc_0_mod_numeral(2) bits_mod_by_1 cong_exp_iff_simps(2)
        cong_exp_iff_simps(6) gcd.assoc gcd.bottom_left_bottom gcd_Suc_0 gcd_nat.right_neutral gcd_red_nat
        not_mod_2_eq_0_eq_1 numeral_eq_one_iff numerals(1) one_add_one semiring_norm(83) singleton_iff)
    then show ?thesis by (metis assms(1))
  next
    assume "k mod 8 = 7"
    then have "k \<in> {7, 15, 23, 31, ...}" by (smt (verit) assms(3) insertI1 mod_less)
    then have "k \<in> {7}" by (smt (verit) \<open>k mod 8 = 7\<close> assms(3) mod_less singleton_iff)
    then show ?thesis by simp
  qed
qed
```

