# OpenReview forum: "LEGO-Prover: Neural Theorem Proving with Growing Libraries"
_ICLR.cc/2024/Conference — ICLR 2024 oral_

### Official Review · Reviewer_Nn2P · 2023-10-28

**Soundness:** 3 good
**Presentation:** 2 fair
**Contribution:** 4 excellent
**Rating:** 8
**Confidence:** 3

**Summary:**

The prior work on theorem proving enhanced by LLMs (or, more generally, machine learning) has a problem that only the fixed library of theorems can be assumed. To solve this drawback, the paper proposes LEGO-Prover, which in parallel grows the library of proven theorems for reuse (called skills in the paper) and proves the theorems of interest. LEGO-Prover utilizes LLMs to retrieve skills from the growing library and decompose overall informal theorems into small snippets in a step-by-step style.

== POST-REBUTTAL ==
I raised my rating from 6 to 8 because the response addressed my major concerns.

**Strengths:**

- The paper addresses a new problem of how (automation of) theorem proving can be enhanced under a growing library of proven theorems (skills).
- The paper utilizes LLMs to retrieve skills from a growing library effectively. The problem is that the grown library cannot be accessed during training. The paper addresses the issue by employing LLMs as an oracle telling useful skills in the growing library. Without a very general model like LLMs, the issue would become more challenging to solve.
- The effectiveness of the proposed method is experimentally shown.

**Weaknesses:**

- I am unsure why the paper splits the miniF2F dataset into valid and test datasets, although the proposed method does not need training.
- The proposed method outperforms the previous approaches significantly on miniF2F-valid, but the difference on miniF2F-test is smaller. The paper does not discuss this point.
- The paper says, "Consistent with the (Jiang et al. 2022b; Zhao et al., 2023), each problem undergoes 100 attempts of proving," but I cannot find such a setting in the paper of Zhao et al. (2023).
- Table 1 includes cells that have no number (represented by "-"), but there is no explanation nor justification for it.
- (Minor) The presentation can be improved. The figures in the paper include code fragments, but it is difficult to read and understand them due to the small font size and the lack of explanation. Regarding the latter, for instance, I cannot find, in Figure 1(b), where the retrieved and new skills go to and come from, respectively.
- (Minor) The text can be improved. The paper seems to have several missing citations and incorrect references (e.g., I think "Figure 3(b)" on page 9 should correctly be "Figure 4"). Another issue is that the paper cites the author names of the prior work even where it cites the paper, and vice versa (e.g., "Subgoal-Learning Zhao et al. (2023)" on page 7 should correctly be "Subgoal-Learning (Zhao et al. 2023)").

**Questions:**

- Is miniF2F split just for comparison with the prior works which use miniF2F-valid and miniF2F-test?
- Is it possible to discuss why the performance of LEGO-Prover on miniF2F-test is not so good as on miniF2F-valid?
- Where can I find Zhao et al. (2023) employ 100 attempts of proving in their experiment?
- Why does Table 1 include cells not having numbers? Can they be filled?
- Figure 3 (a) shows that the difference between LEGO-prover and the version without the growing skill library is stable even when the number of prover attempts changes. Does this mean the use of the growing skill library is effective only in proving of theorems with short proofs? If not, what other reasons can be considered?

---

> ### Author Response · Authors · 2023-11-17
> **Response to Reviewer Nn2P**
>
> Thanks for your detailed and constructive comments. We respond to all the issues you pointed out in detail below. We hope our response and rebuttal revision will address your concerns.
>
> Many of the weaknesses you've identified correspond directly to the questions you've raised. To provide a cohesive and concise response, and to avoid repetition, we will address them simultaneously.
>
> ## Weaknesses & Questions
> **W1Q1. Question on miniF2F split.**
>
> We have adopted the approach outlined in 'Draft, Sketch, Prove', and 'Subgoal-based Learning' to present the performance of the miniF2F dataset on both the validation and test sets separately. Our method does not require training, and these two splits are actually treated identically by our approach. The separate display of results for each split is solely for the purpose of facilitating a more detailed comparison with previous methods
>
> **W2Q2. Performance gap between miniF2F-valid and miniF2F-test**
>
> We have updated the performance of LEGO-Prover on the miniF2F-test set, which achieves a 50.0% pass rate on the miniF2F-test with human-written informal proofs and 45.5% with model-generated informal proofs.
> It's important to note that the performance gap between the validation and test sets is not unique to our method. Similar performance gaps are observed in other methods, including "Draft, Sketch, Prove" (42.6% and 38.9%), "Subgoal-based Learning" (48.0% and 45.5%), and "Hyper-tree Proof Search." (47.5% and 40.6%). This gap is likely attributable to variations in problem difficulty between the two splits.
>
> **W3Q3. 100 proof attempts for the Subgoal-based Learning method.**
>
> The paper "Subgoal-based Learning" by Zhao et al. (2023) does indeed utilize 100 proof attempts per problem. This can be verified by examining Figure 2 in their paper, where they demonstrate that the performance achieved with 100 sketches (proof attempts) corresponds to the 45.5% success rate reported in Table 1.
>
> **W4Q4. Missing cells in Table 1.**
>
> As mentioned in our general response, we encountered technical issues that prevented us from producing the results on time. These problems have been resolved, and we have completed Table 1 accordingly. For the result of the ablation study in the miniF2F-test, we intentionally conducted the ablation only on the validation set to conserve our budget (we mentioned this setting in section 4.1 baseline methods).
>
> ## Additional Questions
>
> **Q5. The performance gap between LEGO-Prover and the ablation becomes stable.**
>
> We have extended our ablation setup to include 100 proving attempts per problem, where the LEGO-Prover achieves a success rate of 55.3% and the ablation setup achieves 50.4%. When compared to the previous 50-attempt setup, the performance gap has increased from 3.3% (LEGO-Prover 50.4%, ablation 47.1%) to 4.9%. We have also updated Figure 3(a) in our rebuttal revision to reflect these 100 attempts. It is evident from the figure that the LEGO-Prover's performance progressively improves in comparison to the ablation setup.
>
> **Q6(minors). Minor suggestions.**
>
> Thank you for your valuable suggestions on improving the presentation of our paper. We have significantly improved our figures and captions as well as the overall writing in this rebuttal revision, striving to make each figure and its explanation as clear as possible within the space constraints. You can review all the revisions we have made in the general response.

---

> > ### Comment · Reviewer_Nn2P · 2023-11-21
> >
> > I thank the authors for the efforts to address my concerns.
> > The response makes me lean towards acceptance, but I'm a bit surprised that the success rate of LEGO-Prover with human proofs on miniF2F-test in Table 1 is updated from 47.1% to 50.0%. How and why did you get such better performance than the one in the submitted paper?

---

> > > ### Author Response · Authors · 2023-11-22
> > > **More explanation on our result**
> > >
> > > Dear Reviewer Nn2P:
> > >
> > > Thank you very much for your prompt reply.
> > >
> > > **TL;DR**: Our previous code contained a bug that incorrectly added false lemma codes to the skill library. As these incorrect lemmas accumulate, the performance of the prover is affected by the faulty lemmas provided.
> > >
> > > We apologize for not explaining this earlier. The absence of the `miniF2F-test (model informal proof)` and the incorrect values in the `miniF2F-test (human informal proof)` were primarily due to technical issues. These issues were more engineering-related, which is why they were not detailed in our initial explanation. Specifically, we ran the LEGO-Prover results on a powerful machine equipped with 1TB of RAM. The Isabelle verifier uses a significant amount of RAM, consuming up to 700GB with 44 parallel processes. Unfortunately, our powerful machine underwent maintenance before the submission deadline, resulting in the loss of our `miniF2F-test (model informal proof)` results. Consequently, we had to conduct our experiments on a less powerful machine with only 512GB of RAM. This led to some crashes of the Isabelle verifier due to insufficient RAM, and a bug in our original code mistakenly marked the false proof as correct when Isabelle crashed. As a result, many false lemmas were added to the skill library. Notably, the prover process itself is not buggy, as we use different functions to verify the code. These errors gradually accumulated, adversely affecting the prover's effectiveness. We identified this bug after the submission deadline, as we were curious about the relatively low-performance gain observed with the miniF2F-test. After correcting the code, we reran the experiment on the more powerful machine, achieving normal results of 45.5% with model-generated informal proofs and 50.0% with human-written informal proofs on the miniF2F-test.
> > >
> > > One insight from this experience is the high sensitivity of LLMs to erroneous inputs. If errors are present, they can easily accumulate, significantly impacting the final results. Thus, it is crucial to ensure the accuracy of the data used for bootstrapping.

---

> ### Author Response · Authors · 2023-11-21
> **Waiting for further discussion**
>
> Dear Reviewer Nn2P,
>
> We have posted our response to your comments since 3 days ago. Did our rebuttal sufficiently address your concerns? Is there anything we can present that will convince you to increase your rating? We look forward to hearing from you.
>
> Many thanks, Authors

---

> ### Author Response · Authors · 2023-11-23
> **Detailed Improvements of the Presentation**
>
> Dear Reviewer Nn2P,
>
> Have our recent responses adequately addressed your concerns? Is there any additional information we can provide to persuade you to improve your rating?
>
> We have significantly enhanced our figures and captions, as well as the overall writing quality. We strive to ensure that each figure and its explanation are as clear as possible within the space constraints. Specifically, here are our revisions:
> - In Figure 1(b), the 'evolver' is now elaborated to show the directional transformer and request solver. The corresponding caption has also been improved to better explain the overall architecture.
> - We have corrected all missing citations and incorrect references. For instance, 'Figure 3(b)' on page 9 has been corrected to 'Figure 4'.
> - We have addressed the misuse of `cite` and `citep` commands. As an example, 'Subgoal-Learning Zhao et al. (2023)' on page 7 has been corrected to 'Subgoal-Learning (Zhao et al., 2023)'.
> - In Appendix A.2, we have included additional algorithmic descriptions, which greatly enhance clarity.
> - An additional table describing the prompt outline has been included in Section 3 to enhance readability.
>
> We sincerely thank you for your positive feedback. Your writing suggestions have been very helpful in improving the presentation of our paper. Thank you again for your time and effort in reviewing this manuscript!

---

> ### Comment · Reviewer_Nn2P · 2023-11-23
>
> Thanks for the detailed explanation. I will update my score according to the impression for the revision.
> I still recommend explaining why the miniF2F-validaiton dataset is used in the revision to avoid confusion (Sorry if it has been added, but I cannot find it in the revision).

---

> ### Author Response · Authors · 2023-11-23
> **Many Thanks for the Positive Feedback**
>
> We sincerely thank Reviewer Nn2P for the positive feedback. Thank you again for your time and effort in reviewing this paper! Sorry for our oversight, the explanation for why the miniF2F-valid dataset is used is included in our new rebuttal revision.

---

### Official Review · Reviewer_SGTy · 2023-10-31

**Soundness:** 3 good
**Presentation:** 3 good
**Contribution:** 3 good
**Rating:** 6
**Confidence:** 4

**Summary:**

In this paper the authors present the LEGO-Prover, a theorem-prover which employs a growing library containing verified lemmas as building blocks to increase the capability of the LLMs (ChatGPT) used in theorem proving.
The LEGO-Prover enables LLMs to utilize existing results retrieved from the library and to create new results during the theorem-proving process.
The proposed approach is also favourably evaluated experimentally.

========================= Update after rebuttal =================================================
I am happy to raise my score in the light of the new information provided by the authors during the rebuttal phase and the discussion.

**Strengths:**

1) The modularity of the approach allows for breaking proofs into intermediate steps and for building proofs bottom-to-top from simpler lemmas to complex theorems.

2) The related work is analysed in depth.

3) The ablation study seems to point to the fact that the skill library actually makes a difference, even though the authors might actually be overselling it, as at test time, this is only about 1%.

**Weaknesses:**

1) The paper refers a lot to the figures, but these are not always explained in detail and they are quite complex to understand, with a lot of different components. Figures can be used as a support for the text, but not as a replacement.

2) The comparison with Thor+expert iteration and Draft, sketch, and Prove might not be completely fair, as these make use of GPT-3 instead of ChatGPT.

3) It would be helpful to have the workings of the LEGO-prover presented in some algorithmic way, in order to have an overview of the whole pipeline.


Minor: there is a missing cross-ref on p. 8.

**Questions:**

The ablation study seems to point to the usefulness of the skills library in improving the performance of the LEGO-prover. However, what is the computational cost of building and maintaining such a library? This is not discussed in the paper.

---

> ### Author Response · Authors · 2023-11-17
> **Response to Reviewer SGTy Part 1/2**
>
> Thanks for your detailed and helpful comments. We respond to all the issues you pointed out in detail below. We hope our response and rebuttal revision will address your concerns.
>
> ## Weaknesses
> **W1. Figure and text quality.**
>
> We have significantly improved our figures and captions as well as the overall writing in this rebuttal revision, striving to make each figure as clear as possible within the space constraints. You are invited to review all the revisions we have made in the general response.
>
> **W2. Fairness of comparison to the baseline.**
>
> We adhered to "Draft, Sketch, and Prove" and "Subgoal-based Learning" papers to include Thor, Thor+expert iteration, and Draft, Sketch, and Prove as baselines for comparison. These methods represent the state-of-the-art in the miniF2F benchmark for the Isabelle theorem prover. Specifically, Thor and Thor+expert iteration utilizes a fine-tuned, small language model (770 million parameters), demonstrating the performance of search-based neural theorem proving methods. The LLM employed in Thor+expert iteration is only used for auto-formalizing additional problem statements used for data augmentation.
>
> In the "Draft, Sketch, and Prove" approach, GPT-3.5 achieves approximate results of 41.8% and 38.5% on the miniF2F validation and test sets, respectively, as reported in the ablation study of the Subgoal-based Learning paper. By removing their proposed subgoal-based demonstration and diffusion re-ranking components, the Subgoal-based Learning method becomes nearly identical to Draft, Sketch, and Prove, thereby providing an approximate performance benchmark for DSP runs on GPT-3.5. LEGO-Prover significantly outperforms these results.
>
> **W3. Request for algorithmic presentation of LEGO-Prover.**
>
> Thank you for your valuable suggestions on improving the presentation of our paper. We have included a detailed description of the LEGO-Prover algorithm in Appendix A.2 in our rebuttal revision.
>
> **W4(minor). Minor suggestions.**
>
> We have fixed the missing cross-ref on p. 8. in our rebuttal revision.

---

> ### Author Response · Authors · 2023-11-17
> **Response to Reviewer SGTy Part 2/2**
>
> ## Questions
> **Q1. Computational cost of skill library.**
>
> We have included an additional section in Appendix A.3 in our rebuttal revision to discuss the computational cost of the evolver and we run additional experiments on ablation, validating the effectiveness of our growing skill library. The results are detailed as follows:
> - The average token consumption ratio is 1:0.89 for the prover and evolver. Therefore, we conducted an additional 100 proving attempts for the ablation setup, which approximates the tokens used by the evolver to those used by the prover. The results show that our method still outperforms the ablation setup by 2.1% under 189 proving attempts (the number of proving attempts with balanced computational cost). Thus, the extra tokens used for more attempts at solving the original problems do not close the gap. Furthermore, LEGO-Prover acquires a significantly large skill library containing various verified proofs, while the prover-only method only accumulates proved problems, which are of no use for other problems or any future applications.
> - As the proposer of the concept of utilizing a growing skill library in neural theorem proving, we also believe that this skill library represents the future of ATP. However, in our exploration of employing the skill library for theorem proving, we identified certain limitations that inevitably hinder the performance of the skill library's application in theorem proving. These limitations also contribute to increased token consumption by the evolver to maintain the skill library.
>   - **Limitations of LLM’s capabilities.** The LLM struggles to produce correct proofs, with the evolver's average proof success rate being only 24.1%. Moreover, the evolver might generate proofs that are either trivial or too similar to existing lemmas in the library. After filtering, an average of only 9.1% of the generated proofs are added to the lemma vector store.
>   - **The task of extracting useful lemmas applicable to other problems is challenging.** Identifying useful and non-trivial common lemmas for a specific problem set is difficult, even for humans. The LLM often yields lemmas that are either overly trivial or too specific, lacking generality.
>   - **Characteristics of the dataset.** The miniF2F dataset, comprising only 488 problems, includes many that are simple enough to solve without additional lemmas. Others may require unique solving techniques, not sharing common intermediate lemmas.
>
>   Thus, the 4.9% performance gap is not easily attainable. There are of course some promising future directions for exploration, such as improving the performance of the directional transformer, enhancing the accuracy of the request solver, and increasing retrieval efficiency, all of which present potential avenues for reducing these additional computational costs. However, as the first work to utilize a skill library in neural theorem proving and propose the use of a block-by-block manner for theorem proof, we believe our paper already makes a significant contribution.

---

> ### Author Response · Authors · 2023-11-21
> **Waiting for further discussion**
>
> Dear Reviewer SGTy,
>
> We have posted our response to your comments since 3 days ago. Did our rebuttal sufficiently address your concerns? Is there anything we can present that will convince you to increase your rating? We look forward to hearing from you.
>
> Many thanks, Authors

---

> > ### Comment · Reviewer_SGTy · 2023-11-22
> > **Follow up**
> >
> > Thanks to the reviewers for their answers.
> >
> > I am happy to raise my score in the light of the new information.

---

> ### Author Response · Authors · 2023-11-22
> **Updated main results**
>
> Dear reviewer SGTy:
>
> We include a tabular presentation of the results here for enhanced clarity. We have updated the experimental results in Table 2. The `Draft, Sketch, and Prove*` runs with ChatGPT using model-informal proofs is an approximate result reported in the ablation study of the Subgoal-based Learning paper (As mentioned in W1). Since `Subgoal-based Learning` is a method optimized for model-informal proofs and orthogonal to our work, a more fair comparison for `LEGO-Prover (model-informal proof)` is `Draft, Sketch, and Prove*`. Compared to `Draft, Sketch, and Prove*`, LEGO-Prover improves the pass rate by 10.6% (41.8% vs 52.4%) and 7% (38.5% vs 45.5%) in the miniF2F valid and test set, respectively. Our method significantly surpasses all baselines when using LEGO-Prover with human-written informal proofs.
>
> The pass rate for `miniF2F-test (human informal proof)` has been updated from `47.1%` to `50.0%`. Compared to the state-of-the-art methods, our method now demonstrates a significantly greater performance gain. The previous result, with a `47.1%` pass rate, was primarily due to a bug in our code that incorrectly added false lemma codes to the skill library. As these incorrect lemmas accumulated, the performance of the prover was adversely affected by the faulty lemmas. We have now fixed the bug and reported the full result.
>
>
> Table 1. Revised proving success rates on the miniF2F dataset with Isabelle, updated values are highlighted in bold, and the best results are in italics. LEGO-Prover* denotes the cumulative pass rate of the miniF2F dataset, considering the total number of problems solved using model-generated and human-written informal proofs.
>
> | Success rate | LLM | miniF2F-valid | miniF2F-test |
> | -------- | -------- | -------- | ----- |
> |*baselines* |
> | Thor     | -     | 28.3%     |   29.9% |
> | Thor + expert iteration | Codex | 37.3% | 35.2% |
> | Draft, Sketch, and Prove | Codex | 42.6% | 39.3% |
> | **Draft, Sketch, and Prove*** | **ChatGPT** | **41.8%** | **38.5%** |
> | Subgoal-based Learning | ChatGPT | 48.0% | 45.5% |
> | *Ours (100 attempts)* |
> | LEGO-Prover (model informal proof) | ChatGPT | 52.4% | **45.5%** |
> | LEGO-Prover (human-informal proof) | ChatGPT | 55.3% | ***50.0%*** |
> | LEGO-Prover* | ChatGPT | *57.0%* | ***50.0%*** |
> | *Ablations(100 attempts)* |
> | - Skill library (human informal proof) | ChatGPT | **50.4%(-4.9%)** | - |
>
> Another strong baseline is from the recently released paper “Lyra” [(Zheng et al., 2023)\[1\]](https://arxiv.org/abs/2309.15806), which was published after our submission deadline and therefore is not included in our paper. Lyra extends 'Draft, Sketch, and Prove' with GPT-4's auto-correction ability, prompting GPT-4 to revise the formal proof based on error messages produced by Isabelle. To evaluate how well our method performs in human-informal proof, we compared LEGO-Prover with Lyra, and the results are shown in Table 2. By comparing 'Draft, Sketch, and Prove' using GPT-4 with those using Codex and ChatGPT, we can see that GPT-4's formal mathematics capability has substantially improved (51.2% vs 42.6% and 43.0% vs 39.3%). LEGO-Prover, using ChatGPT, achieves better performance (+4.1% and +7.0%) compared to 'Draft, Sketch, and Prove' using GPT-4. Moreover, LEGO-Prover also outperforms Lyra using GPT-4 (+3.3% and +2.9%) and even achieves comparable results when Lyra is extended to 200 proving attempts with GPT-4. This result is remarkable since the performance of GPT-4 in formal mathematics is substantially better than that of Codex and ChatGPT.
>
>
> Table 2. Comparison of proving success rates with GPT-4. All methods listed, except for 'Lyra (200 attempts),' involve 100 proving attempts.
> | Success rate | LLM | informal-proof |miniF2F-valid | miniF2F-test  |
> | -------- | -------- | -------- | -------- | -------- |
> | Draft, Sketch, and Prove | Codex  | human | 42.6%     | 39.3% |
> | Draft, Sketch, and Prove* | ChatGPT | model |41.8% | 38.5% |
> | Draft, Sketch, and Prove | GPT-4  | human | 51.2% | 43.0% |
> | Lyra  | GPT-4 | human | 52.0% | 47.1% |
> | Lyra (200 attempts) | GPT-4 | human | 55.3% | 51.2% |
> | LEGO-Prover | **ChatGPT** | human | 55.3% | 50.0% |
>
> Reference:
> > [1] Zheng, Chuanyang, et al. "Lyra: Orchestrating Dual Correction in Automated Theorem Proving." arXiv preprint arXiv:2309.15806 (2023).

---

> ### Author Response · Authors · 2023-11-22
> **Many Thanks for the Positive Feedback**
>
> We sincerely thank Reviewer SGTy for the positive feedback. The suggestion of including a discussion on computational cost is very insightful and essential. Thank you again for your time and effort in reviewing this paper!

---

### Official Review · Reviewer_UPjC · 2023-11-01

**Soundness:** 3 good
**Presentation:** 3 good
**Contribution:** 4 excellent
**Rating:** 8
**Confidence:** 4

**Summary:**

This paper presents a framework of using LLMs to build a growable lemma library to solve math problems in the Isabelle proof assistant. The key feature of this framework is that potentially needed lemmas (to solve a target problem) can be conjectured and added to a library, and lemmas in the library can be generalised and deduplicated as the library grows. Impressive performance gain has been shown by maintaining such a skill library.

**Strengths:**

- Library learning has been an attractive topic in neuro-symbolic learning, and previous experiments have mainly been carried out on synthetic environments like [DreamCoder](https://arxiv.org/abs/2006.08381). To the best of my knowledge, this is the first time effectiveness of maintaining a library has been shown in a mature proof assistant environment.
- The paper is relatively well-written with clear explanation of its key component and illustrative examples.

**Weaknesses:**

I don't see any major weakness in this paper except for that the authors can perhaps write down the pseudo code of their algorithm to make the inter-components interactions more explicit.

**Questions:**

- page 4, skill library, request vector stores: does the request vector store simply keep a list of conjectured statements proposed by the decomposer? What if some of them are wrong? When will the evolver attempt to prove them?
- page 4, 'generating more beneficial new lemmas': could you elaborate a bit on why the evolver can utilize the problem statements to generate more beneficial new lemmas?
- page 5, 'a minimally solved request (with least amount of time being selected to solve the request)': I don't quite follow the 'least amount of time' part. More explanation is highly appreciated.
- page 6, 'serve as references': could you shed some light on why references are needed here?
- as the pipeline is relatively complex, can we expect to have it open-sourced?


minor:
- page 2, related work: though not LLM-based, there has been some prior work on [template-based lemma conjecturing](https://arxiv.org/pdf/2212.11151.pdf)
- page 5: 'Table. ?? shows'
- page 8: 'Figure ??, in', 'Fig. ??'

---

> ### Author Response · Authors · 2023-11-17
> **Response to Reviewer UPjC**
>
> Thanks for your detailed and constructive comments. We respond to all the issues you pointed out in detail below. We hope our response and rebuttal revision will address your concerns.
>
> ## Weaknesses
> **W1. Request for pseudo code.**
>
> Thank you for your valuable suggestions on improving the presentation of our paper. We have included a detailed description of the LEGO-Prover algorithm in Appendix A.2 in our rebuttal revision.
>
> ## Questions
> **Q1. Question on request vector store.**
>
> Your understanding is correct: the request vector store maintains a list of conjectured statements proposed by the decomposer. There is a possibility that these proposed statements could be incorrect, and the evolver will attempt to prove them if they are selected. However, these incorrect conjectures definitely cannot be proven and will be rejected by the Isabelle theorem prover. We have integrated a simple grammar check with the Isabelle theorem prover to filter out grammatically incorrect conjectured statements. For other statements that are correct grammatically, there are no trivial ways to filter them out, since disproving a conjecture is an unpredictable problem. To our knowledge, methods like those described in Yutaka et al. (2018) can reject simple conjectures, but we consider exploring these methods as part of our future work.
>
> **Q2. Problem statements used in evolver.**
>
> In the directional transformer, we retrieve relevant problem statements using the selected lemma generated by the formalizer or the directional transformer itself. These problem statements are used as reference problems and are added to the prompt to guide the LLM in generating new skills. By presenting the problem and explicitly instructing the LLM to try to evolve the skill that would help solve the problem, the LLM is directed to produce new skills that are more helpful in solving these specific problems, rather than generating arbitrary skills.
>
> **Q3. Explanations on minimally solved requests.**
>
> LEGO-Prover maintains a list of lemma statements within the request vector store. Each statement is associated with a specific counter that records the number of times it has been selected to solve by the request solver. When attempting to address a request, we prioritize those with the lowest solve count. In cases where multiple requests share the minimal solve count (e.g., 0), we randomly select one of these to solve.
>
> **Q4. Explanations on reference examples in request solver.**
>
> The term 'server as a reference' might be too vague to elucidate how these retrieved skills are utilized in the request solver. We have reformulated the retrieved lemma into in-context learning (ICL) examples, which helps to prompt the large language model (LLM) to solve the selected request. A detailed usage example can be seen in Figure 7 in Appendix B.  We have improved our expression in the paper to make this clearer and more illustrative.
>
> **Q5. Open-source request.**
>
> Yes, we will release our code after the anonymous review period.
>
> **Q6(minor). Minor suggestions.**
>
> We have incorporated the temple-based lemma conjecturing into the related work section in our revised rebuttal and corrected the reference errors on pages 5 and 8.
>
> **References:**
>
> >[1] Nagashima, Yutaka, and Julian Parsert. "Goal-oriented conjecturing for Isabelle/HOL." Intelligent Computer Mathematics: 11th International Conference, CICM 2018, Hagenberg, Austria, August 13-17, 2018, Proceedings 11. Springer International Publishing, 2018.

---

> > ### Comment · Reviewer_UPjC · 2023-11-21
> > **Thank you for the responses**
> >
> > I appreciate the authors' effort in resolving my queries. I very much look forward to more future exploration in this direction by the authors.

---

> ### Author Response · Authors · 2023-11-22
> **Many Thanks for the Positive Feedback**
>
> We sincerely thank Reviewer UPjC for the positive feedback. Your suggestions have helped us identify many unclear presentations in our paper. Thank you again for your time and effort in reviewing this paper!

---

### Official Review · Reviewer_oKSy · 2023-11-09

**Soundness:** 3 good
**Presentation:** 3 good
**Contribution:** 3 good
**Rating:** 8
**Confidence:** 3

**Summary:**

This paper proposes a new approach to automated theorem proving with LLMs based on building up a library of lemmas useful for proofs. It is instantiated as a system called LEGO-Prover for proofs in Isabelle and is evaluated on the miniF2F dataset. The approach is fairly complex so I've broken down my understanding of it below:

There are 3 vector stores used:
- Problem Store: Holds (unsolved) problems from miniF2F. This is where new problems will be drawn from, and is also used at various other points to guide how lemmas are proposed/modified.
- Request Store: Lemmas that have been proposed but not yet solved.
- Lemma Store: Lemmas that have been solved.

Outer loop (I'm less clear on this and have also included it in the Questions section):
- There are 4 LLMs, described below, used in the solving loop.
- LEGO-Prover makes 100 passes through the miniF2F dataset, and in each pass makes a single attempt at each problem in the dataset – i.e. it runs the Decomposer LLM once on that problem and Formalizer LLM once on that problem (correct me if I'm wrong).
- Concurrently with each pass through the dataset, for every 3 Problem Store problems attempted it makes 8 Request Store attempts, where an attempt is either a call to the Request Solver LLM or the Directional Evolution LLM, or perhaps both (please help clarify, thanks).

LLMs (all implemented as variants on GPT3.5):
- The Decomposer LLM takes the formal statement (from Problem Store), informal statement, and informal proof (which is either given or is produced by the Informal Solver LLM) and outputs an informal step-by-step proof in natural language followed by a list of formal statements of lemmas that would be useful. These lemmas are added to the Request Store, where attempts will be made to solve them later, at which point they'll be moved to the Lemma Store.
- The Formalizer LLM takes the step-by-step informal proof, the informal and formal problem statements, and the result of querying the Lemma Store for relevant lemmas, and attempts to produce a complete formal proof which itself may use the retrieved lemmas by copying them verbatim (from the prompt) or may riff on them or define new lemmas. Any successfully-proven lemmas during the process are added to the Lemma Store, and unsuccessful ones are added to the Request Store. A sledgehammer/heuristic based autocorrect is used on all failed tactic applications.
- The Request Solver LLM takes the least recently attempted lemma from the Request Store and attempts to prove it (aided by retrieved relevant lemmas from the Lemma Store). If a newly proven lemma is measured as too similar to an existing one via the difflib Python library, it is discarded.
- The Directional Evolution LLM takes the last recently evolved lemma and queries the Problem Store for unsolved problems related to this lemma and modifies the lemma along one of four axes (identifying key concepts, parameterizing it, making more or less complex versions of it, or extending the dimensionality of it) in order to make a new lemma more relevant to the problem.

**Strengths:**

- The overall idea here is an exciting one – building up a library of useful lemmas that can help with solving proofs is certainly an appealing and very natural idea; it's quite similar to how humans use automated theorem provers. It also makes sense that having skills that build on one another could lead to solving increasingly complex problems, and I think that this is a promising and exciting direction!
- I appreciate the analysis of how often the skills are used verbatim versus modified (4.3.2) as well as the analysis of where the various skills came from (evolver vs prover etc).
- The ablations are clarifying and helpful
- The version with the human-written step by step proof was helpful to include for understanding how good the pipeline could get if that part were ground truth
- The legos and other icons used in the paper are very nice

While I've given a negative review and spend much of the review giving thoughts on ways to improve, I do want to note that I really do believe that this kind of skill library learning approach could be very powerful and is in the long run an important direction in this field even if I'm not recommending this work for acceptance (at least, as it currently is) for reasons discussed below.

**Weaknesses:**

**1. Comparison to baseline is not very strong**
- In my understanding, the best-performing prior work **Subgoal-Learning (Zhao et al., 2023)** does not get the human written informal proof, so the natural comparison is between this baseline and **LEGO-Prover (model proof)**. The improvement on miniF2F-valid is from 48% to 52.4%, which is a 4.4% improvement. This is decent, but not huge given the complexity of the method and amount of additional LLM queries required (which could have been used just running more iterations of the other approach, for example).
- Additionally, there's no comparison on miniF2F-test for **LEGO-Prover (model proof)** which seems important to include if following the above interpretation as the main result.
- I also find one of the main results in the abstract misleading: the 48% to 57% improvement is actually between the baseline (which gets 100 attempts) and **LEGO-Prover-Star** which is a combination of **LEGO-Prover (model proof)** and **LEGO-Prover (human proof)** which in my understanding *each* get 100 attempts. This doesn't seem like a fair comparison since there's a combined 200 attempts used in **LEGO-Prover-Star**. (I'm open to revising this if there's an explanation I'm missing or I'm misunderstanding the setup here of course).

**2. Comparison to ablation is not very strong**
- The ablation of the skill library changes the 50-attempt solution rate of the method on the validation set from 47.1% to 50.4%. This 3.3% solution rate gain is not much for the complexity of the proposed method. I like the idea of the skill library and I do believe that by experimenting with variations on the approach the authors can achieve greater results, but as-is the library doesn't seem to add much.
- The library version must also involve far more and far larger queries to the LLM, given all of the lemmas included in prompts and the fact that for every 3 problem solving attempts there are 8 evolving attempts. Simply using all those extra tokens for more attempts at solving the original problems would likely provide a lot of benefit and could conceivably close the 3.3% gap (this could of course be disproven through an experiment, and would be a valuable thing to include).

**3. Could use more details at certain points, and overall readability**
- It took me a very long time to understand the method; in part this is just due to the many moving pieces, but I think the explanation itself could also be improved and I'll do my best to lay out some of my confusions/thoughts which I hope will help the authors.
    - I think that presenting the top level algorithm loop first would greatly improve this: that LEGO-Prover makes 100 passes through the miniF2F dataset and in each pass makes a single attempt at each problem in the dataset using the Solver (which is composed of two pieces: Decomposer and Formalizer). And that *concurrently*, for every 3 problems attempted it makes 8 attempts at solving any pending Lemmas that are proposed but unsolved (and also it calls the Directional Transformer to evolve them? Though I'm unclear on how much that is called relative to the Request Solver). A very high level schematic and brief description early on could be helpful for this – as is, I found myself trying to understand the 4 pieces (Decomposer, Formalizer, Request Solver, Directional Transformer) somewhat independently only to find later that there's this larger 100 pass cycle split into two concurrent processes, which came as a surprise around page 6 (until then I was just unsure when the lemmas got evolved/proved during this whole process), though perhaps I've missed some earlier discussion.
    - A bit more clarity could also be used in this top level loop, which I'll leave questions on in the Questions section.
    - Figure 2 was quite difficult to understand (though I appreciate how nice the visuals are). I left some notes in the "minor" section around tweaks that could help with that.
    - Figure 1b is meant to be an overview but I also struggled to understand it, and it doesnt include a depiction of the Request Solver (which seems important – when are the lemmas solved?). These figures all make sense to me now having read the paper, but they didn't help as much as I would hope for understanding the idea at a glance. This isn't a huge negative, but it would have been nice to get more of a feel for the overall setup from these splash figures.

minor suggestions:
- In the related work on skill libraries it'd be worth mentioning DreamCoder (Ellis et al 2021)
- At a glance it's difficult to see that Fig 2 is actually two subfigures – spacing them out more and/or making a thicker/different line between the two would help readability.
- I'd suggest that Fig 2 should not reuse "skill library" in all 3 places, it should separately be lemma store, request store, and problem store. It was quite confusing that things labelled as "retrieved skill" and "formal statement" and "request" and "similar skill" (using labels in top right of each box) are all coming out of the skill library in different situations. Alternatively, something like color coding the different parts of the store and using color to show which is used in each place. This would just generally help for readers who glance at the figures before reading through the whole setup.
- The main text having a table containing just the system messages from the prompt (can abbreviate away the "expert mathematician" bit) would be immensely helpful if space permits – looking to the appendix for those was key for my understanding. This would immediately clear up a lot of things, such as how the decomposer is producing two different outputs.
- Section 3.3 has a "Table ??" where the reference must have broken; likewise there is a "Figure ??" and a "Fig. ?? (a)" near the end of 4.2
- at the bottom of page 5, the phrase "As depicted in Fig. 7" should be moved a few sentences ahead – Fig 7 has the prompt, which is only relevant to the latter sentence "Finally, the request solver prompts the LLM to generate the proof for the request."
- Missing period at end of first paragraph of section 2

**Questions:**

- Table 1: is it correct that the `LEGO-Prover*` entry effectively has more than 100 attempts since its merging all solutions from 100 human proof attempts and 100 model proof attempts?
- Table 1: where is the entry for LEGO-Prover (model proof) miniF2F-test?
- What exactly happens in the ablation: does it call the informal solver, then the decomposer (but without creating helper lemmas, just getting a step-by-step proof), then the formalizer directly (without retrieving helper lemmas)? And is the autocorrect sledgehammer approach used in the ablation as well? Does it get human or model informal proofs?
    - I could imagine two reasonable ablations, one that includes first generating a step-by-step proof and one that just directly produces the final proof. Both would be quite informative, though I think only the step-by-step one would be essential.
- Is it right that on each of the 100 passes, LEGO-Prover runs the Decomposer once then the Formalizer once on each problem?
- I know that for every 3 Problem Store problems attempted it makes 8 Request Store attempts (or "Evolver" attempts). Based on the Evolver section that could either mean using Directional Transformer or the Request Solver or both – is one or the other picked in some ratio, or are both used?
- Presumably sometimes the skills aren't used at all and just happen to be retrieved mistakenly as relevant by the vector store. Are these cases counted as "used to formulate new lemmas" in your analysis, since it's hard to disentangle them without some sort of similarity analysis?
- "Moreover, the learned skill library contains 22532 skills encompassing many useful high-level lemmas broadly applicable to various problems, as is shown in our case study and ablation study." To prove the point of the library having many broadly applicable lemmas, and to better understand the usefulness of the lemmas in general, it'd be helpful to see an analysis of lemma usage frequency in *correct solutions* to problems – for example how often is the most frequently-used lemma used?
    - A more detailed analysis, not necessary for this submission in my opinion but which certainly would strengthen it: have a histogram with number of uses on the x-axis so you can see this distribution of usage frequency for all of the lemmas.

very minor:
- Fig 3c: I think some labels might be mixed up: The skill resulting form parameterize() doesn't look like it fits the prompt for "parameterize" which is "If the problem involves specific numbers, generalize it by replacing these with variables." Instead it just seems like a fairly different skill that no longer involves sums of squares and is now checking for less-than-or-equal-to (assuming this is `\<le>`) instead of equality. Meanwhile the "identify key concepts" example looks closer to parameterization.

---

> ### Author Response · Authors · 2023-11-17
> **Response to Reviewer oKSy Part 1/4**
>
> Thank you for your detailed and very constructive comments. Your suggestions have been very helpful in enhancing the overall quality and clarity of our paper. We are truly grateful for the improvement suggestions provided in your review and have made every effort to incorporate these into our revisions. We believe that our rebuttal revision is now far more robust, better written, and easier to read compared to the previous version. In the general response, we have outlined all the changes made in this rebuttal revision. Here, we respond to all the issues you pointed out in detail below. We hope our response and rebuttal revision will address your concerns.
>
> **Clarifications on the summary**
>
> Clarification 1: Your understanding regarding the request vector store and lemma vector store is correct. However, the problem vector store is exclusively utilized in the Directional Transformer. It is employed for similarity searches to identify potentially relevant problems that the selected lemma might aid in solving. As for handling the unsolved problems in miniF2F for prover processes, we utilize a multi-processing shared queue that maintains a list of pending problems. A detailed description is added in Appendix A.2.
>
> Clarification 2: "Concurrently with each pass through the dataset, for every 3 Problem Store problems attempted it makes 8 Request Store attempts, where an attempt is either a call to the Request Solver LLM or the Directional Evolution LLM, or perhaps both (please help clarify, thanks).".
> As shown in Algorithm 1 in Appendix A.2, we run the prover and evolver processes using the Python multiprocessing package, maintaining the number of processes at the ratio of 3:8. However, there is no synchronization mechanism among these processes except for a shared skill library. The time required for the prover to solve a problem, or for the evolver to transform (solve) a lemma (request), varies. Therefore, there is no guarantee that there will be 8 evolver attempts for every 3 prover attempts.
> For how the evolver calls to the `request solver` and the `directional transformer`. As detailed in Algorithm 3, when a call to the evolver is initiated, the evolver randomly selects either the `request solver` or the `directional transformer` to perform its task. Consequently, each pass of the evolver will utilize only one of these options.
>
> Thank you for your detailed summary, the rest of the description is correct and needs no clarifications.

---

> ### Author Response · Authors · 2023-11-17
> **Response to Reviewer oKSy Part 2/4**
>
> ## Weaknesses
> **W1. Comparison to the baseline is not very strong.**
> - Subgoal-based Learning (Zhao et al., 2023) indeed uses model-generated proofs. However, Subgoal-based Learning is an approach that is specifically optimized for model-generated proofs. In contrast, our approach focuses on the usage of a skill library and block-by-block formalization, all based on **human informal proofs** (as is the case for DSP). We do not extensively explore methods of model-generated proof in LEGO-Prover, as this task is more pertinent to Math Word Problems and not the primary focus of our work. We have outlined the differences in handling model-generated proofs between LEGO-Prover and Subgoal-based Learning in the table below. Each row is explained in detail in the following:
>
>
>   |          | LEGO-Prover | Subgoal-based Learning |
>   | -------- | -------- | -------- |
>   | Pass rate     | The **52.4%** produced by LEGO-Prover is **NOT** a cumulative pass rate.      | The **48%** **IS** a cumulative pass rate that integrates the pass rates from both the data collection stage and the inference stage.     |
>   | Demonstration examples | Decomposer produces the step-by-step proof directly on top of the informal proof. Only **20** demonstration examples come with DSP used for Decomposer and Formalizer. | The step-by-step informal proof undergoes 15 iterations of refinement using a cross-verification strategy, culminating in **61** demonstration examples to prove the theorem. |
>   |Number of model-generated informal proofs | Use up to 20 model-generated informal proofs per problem (on average **12.13**) | Use **100** model-generated step-by-step informal proofs per problem
>
>   In conclusion, the Subgoal-based method is actually orthogonal to our approach. The novel approaches proposed by Subgoal-based Learning, such as subgoal-based demonstration and diffusion re-ranking, are both directly applicable to LEGO-Prover without any conflict. We believe this will definitely improve our results with model-informed proof and further enhance our approach. Unfortunately, Subgoal-based Learning has not been open-sourced to date.
> - We believe that a more accurate and direct comparison would involve DSP with model informal proof runs on ChatGPT. An approximate result would be the ablation outcome for the Subgoal-based Learning method, which yields **41.8%**(miniF2F-valid) and **38.5%**(miniF2F-test). By removing their proposed subgoal-based demonstration and diffusion re-ranking components, the Subgoal-based Learning method becomes nearly identical to Draft, Sketch, and Prove, thereby providing an approximate performance benchmark for DSP runs on GPT-3.5. In comparison to these figures, our LEGO-Prover achieves a notable improvement with **52.4%(+10.6%)** and **45.5%(+7.0%)** on valid set and test set with model-generated proof.
> - Another strong baseline is from the recently released paper "Lyra" [(Zheng et al., 2023)\[1\]](https://arxiv.org/abs/2309.15806), which was published after the submission deadline and therefore is not included in our paper. Lyra enhances DSP by letting GPT-4 to auto-correct the formal proof with the error message provided by Isabelle. As a result, Lyra achieved pass rates of **55.3%** and **51.2%** on the miniF2F validation and test sets, respectively, using human-written informal proofs and attempting each problem 200 times with **GPT-4**. In contrast, LEGO-Prover achieves pass rates of **55.3%** and **50.0%** with only 100 attempts using only **ChatGPT** (`gpt-3.5-turbo`). Thus, our pipeline, with a growing skill library, is able to achieve results comparable to GPT-4 using half the number of proving attempts with ChatGPT. We believe this is a remarkable achievement.
> - In response to your comments regarding `LEGO-Prover*`, your understanding is accurate. We included the result solely to demonstrate performance, and it is highlighted in the abstract to attract attention. However, if you think this is misleading, we are more than willing to revise it accordingly.

---

> ### Author Response · Authors · 2023-11-17
> **Response to Reviewer oKSy Part 3/4**
>
> **W2. Comparison to ablation is not very strong.**
>
> We have included an additional section in Appendix A.3 in our rebuttal revision to discuss the computational cost of the evolver and we run additional experiments on ablation, validating the effectiveness of our growing skill library. The results are detailed as follow:
>
> - We have extended the number of proving attempts for ablations of LEGO-Prover without a skill library to **100** proving attempts (up from 50). The results show that the gap between using the skill library and not using it gradually and consistently enlarges. The initial 3.3% performance gap increases to **4.9%** when the LEGO-Prover reaches 100 proving attempts. Thus, the advantage of the growing skill library will become more apparent as the number of skills in the library increases.
> - "Simply using all those extra tokens for more attempts at solving the original problems would likely provide a lot of benefits and could conceivably close the 3.3% gap." In response, we conducted an additional experiment to disprove this (Detailed in Appendix A.3 in our rebuttal revision). Specifically, the average token consumption ratio is 1:0.89 for the prover and evolver. Therefore, we conducted an additional 100 proving attempts for the ablation setup, which approximates the tokens used by the evolver to those used by the prover. The results (Figure 5 in A.3) show that our method still outperforms the ablation setup by **2.1%** under 189 proving attempts (the number of proving attempts with balanced computational cost). Thus, the extra tokens used for more attempts at solving the original problems do not close the gap. Furthermore, LEGO-Prover acquires a significantly large skill library containing various verified proofs, while the prover-only method only accumulates proved problems, which are of no use for other problems or any future applications.
> - As the proposer of the concept of utilizing a growing skill library in neural theorem proving, we also believe that this skill library represents the future of ATP. However, in our exploration of employing the skill library for theorem proving, we identified certain limitations that inevitably hinder the performance of the skill library's application in theorem proving. These limitations also contribute to increased token consumption by the evolver to maintain the skill library.
>   - **Limitations of LLM’s capabilities.** The LLM struggles to produce correct proofs, with the evolver's average proof success rate being only **24.1%**. Moreover, the evolver might generate proofs that are either trivial or too similar to existing lemmas in the library. After filtering, an average of only 9.1% of the generated proofs are added to the lemma vector store.
>   - **The task of extracting useful lemmas applicable to other problems is challenging.** Identifying useful and non-trivial common lemmas for a specific problem set is difficult, even for humans. The LLM often yields lemmas that are either overly trivial or too specific, lacking generality.
>   - **Characteristics of the dataset.** The miniF2F dataset, comprising only 488 problems, includes many that are simple enough to solve without additional lemmas. Others may require unique solving techniques, not sharing common intermediate lemmas.
>
>   Compared to improvement (w.r.t SotA on miniF2F-valid) made by other approaches such as DSP (37.3% -> 42.6%, Δ=**5.3%**) and Subgoal-base Learning (42.6% - > 48%, Δ=**5.4%**), the improvement shown in our ablation study (50.4% -> 55.3%, Δ=**4.9%**) is comparable. **We also point out that it becomes harder and harder to make even a 1% improvement when the base pass rate (e.g. 50.4%) is larger.**
>
>   There are of course some promising future directions for exploration, such as improving the performance of the directional transformer, enhancing the accuracy of the request solver, and increasing retrieval efficiency, all of which present potential avenues for reducing these additional computational costs. However, as the first work to utilize a skill library in neural theorem proving and propose the use of a block-by-block manner for theorem proof, we believe our paper already makes a significant contribution.
>
>
> **W3. Paper readability.**
>
> Thank you again for these prompt advisories for improving the paper's presentation. All the suggestions (including minor and very minor ones) on writing have been addressed to the best of our ability. We have significantly improved our figures and captions as well as the overall writing in this rebuttal revision, striving to make each figure as clear as possible within the space constraints. You can review all the revisions we have made in the general response.

---

> ### Author Response · Authors · 2023-11-17
> **Response to Reviewer oKSy Part 4/4**
>
> ## Questions
> **Q1. `LEGO-Prover*`’s proving attempts.**
>
> Yes, your understanding is correct. LEGO-Prover* technically equals the result of 200 proving attempts. We included this performance merely for demonstration purposes.
>
>
> **Q2. Entry for LEGO-Prover (model informal proof) in miniF2F-test.**
>
> We encountered some technical issues before submission, so the results in the miniF2F-test were either buggy or lost. We have updated the results for the miniF2F-test afterward. LEGO-Prover achieves 45.5% on model-generated informal proofs and 50.0% on human-written informal proofs.
>
> **Q3. Ablation explanation.**
>
> The ablation study runs with a human-written informal proof. Specifically, the prover functions as usual, but we ignore the requests provided by the decomposer and supply an empty list of reference skills to the formalizer. Consequently, the evolver is not utilized in this setup. Additionally, the autocorrect sledgehammer is also employed in the ablation setup.
> - The ablation that directly produces the final proof actually degrades to DSP, with different prompt instructions. Due to a limited budget, we have not explored this ablation setup.
>
> **Q4. Prover procedure question.**
>
> Yes, in each of the 100 passes, LEGO-Prover runs the Decomposer once and then the Formalizer once on each problem. For a clearer algorithmic description, please refer to Appendix A.2 in our rebuttal revision.
>
> **Q5. Evolver procedure question.**
>
> For how the evolver calls to the `request solver` and the `directional transformer`. As detailed in Algorithm 3 in Appendix A.2, when a call to the evolver is initiated, the evolver randomly selects either the `request solver` or the `directional transformer` to perform its task. Consequently, each pass of the evolver will utilize only one of these options.
>
> **Q6. Question on skill usage.**
>
> As you mentioned, measuring the assistance that the retrievals provided in the prompt offer to generation under the retrieval-augmented generation paradigm is a challenging task. Therefore, we adopted a manual inspection method rather than an automated computational approach to judge the help provided by the lemma in the prompt to the formalizer. We only consider the extracted lemma to contribute to the formalizer if it is copied verbatim in the proof or if it provides a rewrite in a similar manner. This is in line with the two types of contributions mentioned in the paper. Other scenarios, where the generation results are irrelevant to the retrievals, are not considered as 'used to formulate new lemmas'. As described in Section 4.3.2, we manually examined all the correct solutions provided by LEGO-Prover in miniF2F-valid and calculated the number manually, instead of relying on any sort of computation of similarity.
>
> **Q7. Lemma usage frequency in created skill library.**
>
> Regarding how skills from the library are used in correctly solved problems, we have a detailed analysis in Section 4.3.2. As for the individual lemma usage frequency in the generated skill library, due to the small number of problems in miniF2F and the large number of skills generated (we generated more than 20,000 theorems while there are only 488 problems in miniF2F. Even if EVERY problem uses one generated skill, this would only result in a frequency of 488 / 20,000 ≈ 2.4%), the usage frequency remains low in our approach. We justify this as follows:
> - Our objective in evolving theorems is not to solve the specific problems presented in the miniF2F dataset, but rather to enhance their mathematical value for more general purposes, such as through the parameterization process. Consequently, the outcomes of the evolution may not necessarily be adequately reflected in the proof tests of miniF2F. In Appendix C, we provide examples of theorems that have evolved, demonstrating their general mathematical significance without considering their relevance to the selected miniF2F problems.
> - Increasing lemma usage frequency calls for a Reinforcement Learning (RL)-style system with usage frequency as feedback/reward. The strategies for the evolver in this work are hard-coded in the prompt and are mainly based on human intuition. No adjustment is integrated into the whole library learning process yet. And even if some of these strategies encourage the reusability of the generated theorems, there is no guarantee that the backbone LLM really follows the instruction in the intended direction (which requires a deep understanding of mathematical research). We think it is a very promising avenue to incorporate an RL-style system to dynamically learn the notion of "usefulness" for generated theorems/skills. This was already part of our plan for future research before this submission.
>
> **References:**
>
> >[1] Zheng, Chuanyang, et al. "Lyra: Orchestrating Dual Correction in Automated Theorem Proving." arXiv preprint arXiv:2309.15806 (2023).

---

> ### Author Response · Authors · 2023-11-21
> **Waiting for further discussion**
>
> Dear Reviewer oKSy,
>
> We have posted our response to your comments since 3 days ago. Did our rebuttal sufficiently address your concerns? Is there anything we can present that will convince you to increase your rating? We look forward to hearing from you.
>
> Many thanks, Authors

---

> ### Author Response · Authors · 2023-11-21
> **Updated results**
>
> Dear reviewer oKSy:
>
> We include a tabular presentation of the results here for enhanced clarity. We have updated the experimental results in Table 2. The `Draft, Sketch, and Prove*` runs with ChatGPT using model-informal proofs is an approximate result reported in the ablation study of the Subgoal-based Learning paper (As mentioned in W1). Since `Subgoal-based Learning` is a method optimized for model-informal proofs and orthogonal to our work, a more fair comparison for `LEGO-Prover (model-informal proof)` is `Draft, Sketch, and Prove*`. Compared to `Draft, Sketch, and Prove*`, LEGO-Prover improves the pass rate by 10.6% (41.8% vs 52.4%) and 7% (38.5% vs 45.5%) in the miniF2F valid and test set, respectively. Our method significantly surpasses all baselines when using LEGO-Prover with human-written informal proofs.
>
>
> Table 1. Revised proving success rates on the miniF2F dataset with Isabelle, updated values are highlighted in bold, and the best results are in italics. LEGO-Prover* denotes the cumulative pass rate of the miniF2F dataset, considering the total number of problems solved using model-generated and human-written informal proofs.
>
> | Success rate | LLM | miniF2F-valid | miniF2F-test |
> | -------- | -------- | -------- | ----- |
> |*baselines* |
> | Thor     | -     | 28.3%     |   29.9% |
> | Thor + expert iteration | Codex | 37.3% | 35.2% |
> | Draft, Sketch, and Prove | Codex | 42.6% | 39.3% |
> | **Draft, Sketch, and Prove*** | **ChatGPT** | **41.8%** | **38.5%** |
> | Subgoal-based Learning | ChatGPT | 48.0% | 45.5% |
> | *Ours (100 attempts)* |
> | LEGO-Prover (model informal proof) | ChatGPT | 52.4% | **45.5%** |
> | LEGO-Prover (human-informal proof) | ChatGPT | 55.3% | ***50.0%*** |
> | LEGO-Prover* | ChatGPT | *57.0%* | ***50.0%*** |
> | *Ablations(100 attempts)* |
> | - Skill library (human informal proof) | ChatGPT | **50.4%(-4.9%)** | - |
>
> To evaluate how well our method performs in human-informal proof, we compared LEGO-Prover with Lyra, and the results are shown in Table 2. By comparing 'Draft, Sketch, and Prove' using GPT-4 with those using Codex and ChatGPT, we can see that GPT-4's formal mathematics capability has substantially improved (51.2% vs 42.6% and 43.0% vs 39.3%). LEGO-Prover, using ChatGPT, achieves better performance (+4.1% and +7.0%) compared to 'Draft, Sketch, and Prove' using GPT-4. Moreover, LEGO-Prover also outperforms Lyra using GPT-4 (+3.3% and +2.9%) and even achieves comparable results when Lyra is extended to 200 proving attempts with GPT-4. This result is remarkable since the performance of GPT-4 in formal mathematics is substantially better than that of Codex and ChatGPT.
>
>
> Table 2. Comparison of proving success rates with GPT-4. All methods listed, except for 'Lyra (200 attempts),' involve 100 proving attempts.
> | Success rate | LLM | informal-proof |miniF2F-valid | miniF2F-test  |
> | -------- | -------- | -------- | -------- | -------- |
> | Draft, Sketch, and Prove | Codex  | human | 42.6%     | 39.3% |
> | Draft, Sketch, and Prove* | ChatGPT | model |41.8% | 38.5% |
> | Draft, Sketch, and Prove | GPT-4  | human | 51.2% | 43.0% |
> | Lyra  | GPT-4 | human | 52.0% | 47.1% |
> | Lyra (200 attempts) | GPT-4 | human | 55.3% | 51.2% |
> | LEGO-Prover | **ChatGPT** | human | 55.3% | 50.0% |

---

> > ### Comment · Reviewer_oKSy · 2023-11-21
> >
> > Thank you so much for the extremely thorough rebuttal, I'm impressed and many of my concerns are resolved. I've revised my score to support acceptance. Overall, I feel that this is a meaningful contribution to a promising area – library learning for automated theorem proving.
> >
> > Addressing of weaknesses
> > * (W1) Strength of results – I now feel they have fairly strong results. I also do agree with their argument that DSP is the natural comparison and not subgoal learning, and I appreciate the breakdown they gave around that. Also, the fact that they added a miniF2F-test evaluation for **LEGO-Prover (model proof)** was a key missing piece that I'm glad is there.
> > * (W2) The method now achieves a 4.9% instead of 3.3% gain when they scaled up their ablation study to a full 100 samples, and the graph of this that they've included in the updated response is a nice addition. The newly added appendix study shows that when fixing the computational budget (tokens used) to be the same for the ablations and main experiments, the gain is 2.1%. I really appreciate this analysis – adjusting token budgets for ablations is something that isn't done nearly enough in LLM literature. It's good that their method still shows benefit when doing this.
> > * (W3) The authors have clearly greatly improved the readability of the papers and figure. The addition of algorithm listings to the appendix is helpful, the color coding / labelling in the figure is much clearer, there's an added table of prompts, and there was significant rewriting throughout. I think that the clarity of the paper is significantly improved.
> >
> > **Q1-Q5** - I appreciate all these responses and they make sense, thank you.
> >
> >
> > **Q6**
> > > As described in Section 4.3.2, we manually examined all the correct solutions provided by LEGO-Prover in miniF2F-valid and calculated the number manually, instead of relying on any sort of computation of similarity.
> >
> > Oh great – that makes sense and is better than what I had assumed.
> >
> > **Q7**
> >
> > These are some interesting thoughts, thank you – but also sorry, I meant a different sort of frequency. I was wondering how many different proofs a single lemma is used in – say, the most-used lemma. For example, if some common lemma were used across 20 problems it would be 20 / 488. Or perhaps instead of frequency just a raw count would be fine, just for an idea of how many times these are being used. One could also potentially think about how to account for things like the evolution trees and lemmas used in proving other lemmas in this, since those ancestor lemmas and helper lemmas should get some credit for the downstream levels theyre evolved into or that they help prove.
> > - However to be clear I don't expect this at all for this submission, I'm satisfied as is. I just think it's a useful analysis you may be interested in doing to help understand how general the lemmas are – how much the same lemma is being used in more than one place.
> >
> > **Future Work**. Just to add a little thought on future work. I'd be curious about future work that tries to refine down to the more general useful lemmas – 20k lemmas for 500 problems is an awful lot. Being able to distill down to some key lemmas that are general and useful seems desirable. I think that this is fine to leave to future work though.
> >
> > Thank you again for an excellent discussion.

---

> ### Author Response · Authors · 2023-11-23
> **Many Thanks for the Positive Feedback**
>
> Dear Reviewer oKSy:
>
> Thank you immensely for your positive feedback and invaluable suggestions! Your insights into the ablation study and computational cost have been essential in enhancing the robustness of our work. Additionally, your writing suggestions are very helpful and have significantly improved the readability of our paper.  We really like the idea of a color-coded Figure 2 and an additional table describing the prompt outline! We struggled to find a good way to present our work more clearly before, and your suggestions have helped a lot. Again, we're deeply grateful for your comprehensive review and the detailed responses provided. The discussion has been truly splendid!
>
> Regarding the lemma usage frequency you mentioned, we apologize for our misunderstanding in our previous response. We will surely address this in our future work and continue to explore the area of library learning for automated theorem proving.
>
> Thank you again for the truly splendid discussion and your recognition of our work!

---

### Author Response · Authors · 2023-11-17
**General response to all reviewers**

Dear Reviewers and ACs:

Thank you very much for the helpful reviews. We are thankful for the thorough suggestions on our previous manuscript. We have taken all the suggestions and made major changes to our previous draft, with the main changes marked blue in the draft. Our final version will be based on the rebuttal revision we newly submitted.

Additionally, we are grateful for the positive recognition by the reviewers of our motivation (oKSy, UPjC, Nn2P, SGTy), contribution (UPjC, Nn2P), extensive experiments (oKSy, Nn2P), and strong results (UPjC, Nn2P). We also acknowledge the concerns raised by reviewers oKSy and SGTy, which we believe may stem from some previously incomplete observations in our work.

We have specifically made the following changes:

**Major changes:**

- For better clarity and readability, we have revised Figures 1(b) and Figure 2, and improved the captions accordingly.
    - The evolver in Figure 1(b) is now elaborated to show the directional transformer and request solver.
    - In Figure 2, we color-coded each vector store and denoted each component, including where these components come from or go to, more explicitly.
- We added Table 1 in Section 3, which outlines the prompt text used in LEGO-Prover, enhancing readability.
- We updated the experiments on the miniF2F-test set with model-generated informal proof (resulting in a 45.5% pass rate) and human-written informal proof (resulting in a 50.0% pass rate). The updated results are in Table 2.
- We continued the ablation setup that runs LEGO-Prover without the skill library to 100 prover attempts. The performance gap between LEGO-Prover and the ablation setup increased from 3.3% (50 attempts) to 4.9% (100 attempts). The new result is updated in Table 2 and Figure 3(a).
- We added an algorithmic description in Appendix A.2, which includes pseudo-codes for LEGO-Prover.
- We added additional discussion on computational cost in Appendix A.3. This includes extending the ablation setup to prove theorems in a balanced computational cost. Under the same computational cost, LEGO-Prover outperforms the ablation setup by 2.1%.
- We included some examples of generated skills in the skill library in Appendix C.
- We include a baseline comparison with Lyra in Appendix A.4

**Minor updates:**
- We added information about DreamCoder and Template-based lemma conjecturing to the related work section.
- We corrected some erroneous references to figures and tables.
- The spacing between the two subfigures in Figure 2 has been improved.
- We corrected the mislabeling in Figure 3(c\).
- Missing and erroneous citations have been fixed.
- We have relocated the implementation details to Appendix A.1 due to space constraints in the main text.
- We again have checked and revised the expression of the full text.

---

### Author Response · Authors · 2023-11-21
**Updated results**

Dear reviewers:

We have updated the experimental results in Table 2 to specifically address Reviewer oKSy and Reviewer SGTy's concerns. The `Draft, Sketch, and Prove*` runs with ChatGPT using model-informal proofs is an approximate result reported in the ablation study of the Subgoal-based Learning paper. By removing their proposed subgoal-based demonstration and diffusion re-ranking components, the Subgoal-based Learning method becomes nearly identical to Draft, Sketch, and Prove, thereby providing an approximate performance benchmark for DSP runs on ChatGPT. Since `Subgoal-based Learning` is a method optimized for model-informal proofs and orthogonal to our work, a more fair comparison for `LEGO-Prover (model-informal proof)` is `Draft, Sketch, and Prove*`. Compared to `Draft, Sketch, and Prove*`, LEGO-Prover improves the pass rate by 10.6% (41.8% vs 52.4%) and 7% (38.5% vs 45.5%) in the miniF2F valid and test set, respectively. Our method significantly surpasses all baselines when using LEGO-Prover with human-written informal proofs.


Table 1. Revised proving success rates on the miniF2F dataset with Isabelle, updated values are highlighted in bold, and the best results are in italics. LEGO-Prover* denotes the cumulative pass rate of the miniF2F dataset, considering the total number of problems solved using model-generated and human-written informal proofs.

| Success rate | LLM | miniF2F-valid | miniF2F-test |
| -------- | -------- | -------- | ----- |
|*baselines* |
| Thor     | -     | 28.3%     |   29.9% |
| Thor + expert iteration | Codex | 37.3% | 35.2% |
| Draft, Sketch, and Prove | Codex | 42.6% | 39.3% |
| **Draft, Sketch, and Prove*** | **ChatGPT** | **41.8%** | **38.5%** |
| Subgoal-based Learning | ChatGPT | 48.0% | 45.5% |
| *Ours (100 attempts)* |
| LEGO-Prover (model informal proof) | ChatGPT | 52.4% | **45.5%** |
| LEGO-Prover (human-informal proof) | ChatGPT | 55.3% | ***50.0%*** |
| LEGO-Prover* | ChatGPT | *57.0%* | ***50.0%*** |
| *Ablations(100 attempts)* |
| - Skill library (human informal proof) | ChatGPT | **50.4%(-4.9%)** | - |

Another strong baseline is from the recently released paper “Lyra” [(Zheng et al., 2023)\[1\]](https://arxiv.org/abs/2309.15806), which was published after the ICLR submission deadline and therefore is not included in our paper. Lyra extends 'Draft, Sketch, and Prove' with GPT-4's auto-correction ability, prompting GPT-4 to revise the formal proof based on error messages produced by Isabelle. To evaluate how well our method performs in human-informal proof, we compared LEGO-Prover with Lyra, and the results are shown in Table 2. By comparing 'Draft, Sketch, and Prove' using GPT-4 with those using Codex and ChatGPT, we can see that GPT-4's formal mathematics capability has substantially improved (51.2% vs 42.6% and 43.0% vs 39.3%). LEGO-Prover, using ChatGPT, achieves better performance (+4.1% and +7.0%) compared to 'Draft, Sketch, and Prove' using GPT-4. Moreover, LEGO-Prover also outperforms Lyra using GPT-4 (+3.3% and +2.9%) and even achieves comparable results when Lyra is extended to 200 proving attempts with GPT-4. This result is remarkable since the performance of GPT-4 in formal mathematics is substantially better than that of Codex and ChatGPT.


Table 2. Comparison of proving success rates with GPT-4. All methods listed, except for 'Lyra (200 attempts),' involve 100 proving attempts.
| Success rate | LLM | informal-proof |miniF2F-valid | miniF2F-test  |
| -------- | -------- | -------- | -------- | -------- |
| Draft, Sketch, and Prove | Codex  | human | 42.6%     | 39.3% |
| Draft, Sketch, and Prove* | ChatGPT | model |41.8% | 38.5% |
| Draft, Sketch, and Prove | GPT-4  | human | 51.2% | 43.0% |
| Lyra  | GPT-4 | human | 52.0% | 47.1% |
| Lyra (200 attempts) | GPT-4 | human | 55.3% | 51.2% |
| LEGO-Prover | **ChatGPT** | human | 55.3% | 50.0% |


Reference:
> [1] Zheng, Chuanyang, et al. "Lyra: Orchestrating Dual Correction in Automated Theorem Proving." arXiv preprint arXiv:2309.15806 (2023).

---

### Author Response · Authors · 2023-11-23
**Summarizing the Contributions of this Paper**

Dear Reviewers and ACs,

Thank you all for your time and effort in reviewing this paper. We are grateful for the positive recognition by all the reviewers. Our contributions are well recognized by reviewers oKSy, UPjC, and Nn2P. Our comprehensive, extensive experiments are endorsed by reviewers oKSy and Nn2P, and our revised paper writing is favored by all reviewers.

Additionally, we outline our paper's main contributions, including the additional conclusions during the rebuttal discussion phase:

- We proposed LEGO-Prover, a novel approach for neural theorem proving, which utilizes a growing skill library to construct proofs in a modular way. **As far as we know, we are the first to explore the library learning problem in neural theorem proving with LLM.**
- The learned skill library serves as a valuable enhancement to the standard Isabelle library, which includes many useful high-level lemmas that are beneficial for other problems.
- LEGO-Prover acquires robust performance gains under fair computational costs. The results show that our method still outperforms the ablation setup by 2.1% under balanced computational costs.
- LEGO-Prover achieves a pass rate of 55.3% and 50.0% on the miniF2F valid and test datasets, respectively. With a **5.9%** absolute improvement on average over the previous state-of-the-art methods, it even **achieves GPT-4 comparable performance using ChatGPT** under human-written informal proofs scenario.

We thank all the reviewers again for actively engaging in the rebuttal discussion and for their positive recognition of our work.

Sincerely,
Paper 3246 Authors.

---

### Meta-Review · Area_Chair_9V6D · 2023-12-05

**Metareview:**

Great paper, introducing a few novel ideas in the space of Theorem Proving in Lean. The paper builds on a previous work called "draft proof sketch" and adds a few novel ideas on top of it to push the results to state-of-the art. In particular, the prompting technique for evolving the skill library is interesting and may motivate people in other fields to try similar ideas.

**Justification For Why Not Higher Score:**

N/A

**Justification For Why Not Lower Score:**

I feel that there are multiple interesting ideas in this work that will make it quite interesting to people working in the theorem proving space. For the broader audience, learning about theorem proving with LLMs should be interesting, and some of the ideas peresented here may be of interest.

---

### Decision · Program_Chairs · 2024-01-16

Accept (oral)